

# Long-term measurements (2009–2015) of non-methane hydrocarbons (NMHCs) in a megacity of China: implication for emission validation and source control

Yarong Peng[1,2], Hongli Wang[2,*], Qian Wang[2], Shengao Jing[2], Jingyu An[2], Yaqin Gao[2], Cheng Huang[2,*], Rusha Yan[2], Haixia Dai[2], Tiantao Cheng[3,1,5,*], Qiang Zhang[4], Meng Li[4], Li Li[2], Shengrong Lou[2], Shikang Tao[2], Qinyao Hu[2], Jun Lu[2], Changhong Chen[2]

[1] Shanghai Key Laboratory of Atmospheric Particle Pollution and Prevention, Department of Environmental Science and Engineering, Institute of Atmospheric Sciences, Fudan University, Shanghai, 200438, China

[2] State Environmental Protection Key Laboratory of Formation and Prevention of Urban Air Pollution Complex, Shanghai Academy of Environmental Sciences, Shanghai, 200233, China

[3] Department of Atmospheric and Oceanic Sciences, Institute of Atmospheric Sciences, Fudan University, Shanghai, 200438, China

[4] Ministry of Education Key Laboratory for Earth System Modeling, Department of Earth System Science, Tsinghua University, 15  Beijing 100084, China

[5] Big Data Institute for Carbon Emission and Environmental Pollution, Fudan University, Shanghai, 200438, China

*Corresponding author:* Hongli Wang (wanghl@saes.sh.cn), Cheng Huang (huangc@saes.sh.cn) and Tiantao Cheng (ttcheng@fudan.edu.cn)



**Abstract**.

Long term measurements of air pollutants represented the footprints of emissions to some extent, which could provide useful and consecutive evolution of emissions. Both atmospheric concentrations and emissions of many air pollutants have been reported decreasing in the past decade due to the implement of various control measures in China, which were different for non-methane hydrocarbons (NMHCs) with increasing emissions as reported previously. The present study employed the long-term (2009–

2015) NMHCs measurements as well as the related social and economic activities data in Shanghai, a megacity in eastern China, to explore the evolution of NMHCs emissions during the periods. The meteorology and photochemistry which might impact the NMHCs measurements were tested as negligible effects on an annual scale. As a result, NMHCs mixing ratio showed no statistic interannual changes, of which compositions varied significantly. This resulted a statistically decreasing trend of ozone formation potential by 3.8% $yr^{-1}$ ($p < 0.05$, the same below), which should benefit to ozone pollution mitigation as its formation in VOC-

control region in Shanghai. Observed alkanes, aromatics and acetylene increased/decreased by +3.8% $yr^{-1}$, -6.0% $yr^{-1}$ and -7.8% $yr^{-1}$ ($p < 0.05$), respectively, whereas alkenes showed no apparent trend. NMHCs sources were apportioned by the positive matrix factorization (PMF) model. Accordingly, vehicular emissions (-6.5% $yr^{-1}$) and petrochemical industry (-9.4% $yr^{-1}$) presented a significant decreasing trend, but the decrease slowed down in recent years; some positive reductions in solvent usage (-8.0% $yr^{-1}$) appeared after 2010; however, emissions of natural gas (+9.2% $yr^{-1}$) and fuel evaporation (with an increasing fraction) became

more important. Discrepancies between measurements and emission inventory, both in interannual trend and speciation as well as source contributions, emphasized the need for further validation in NMHCs emission inventory in future. Our study confirmed the effectiveness of strengthening emission standards of vehicles and key industrial sources, generally mobile and point sources, against the increasing vehicle populations and industries in the past. While, limited reductions were observed for the fugitive emissions like petrochemical industry which need more effective measures in future. Emissions of natural gas and fuel

evaporation deserved more attention as the quick increase of natural gas and fuel consumption due to the policy of energy structure reformation in China.





# 1 Introduction

With the implementation of a series of clean air action in China, declines in many air pollutants both in ambient concentration and emissions were reported widely, while, worsened surfaced ozone pollution occurred meantime (Ma et al., 2016;Sun et al., 2016b;Xu et al., 2019;Shen et al., 2019). The reported increasing emissions of anthropogenic NMHCs in China (Li et al., 2019f) was recognized as an important role in the worse of ozone pollution, which was attributed to the absence of effective controlling measures of NMHCs. It seemed not consistent with the reality.

Long measurements of air pollutants in ambient could provide useful and consecutive evolution of emissions, which have been used to validate the emissions. As reported, contrast to the growth (28 %) in inventory, measured NMHCs at an urban site in Beijing decreased by 37 % during August from 2004 to 2012 (Wang et al., 2015), and results of source apportionment also showed an inconsistency in trends of solvent use and industry with inventory. Similarly, the trends of both traffic-related and solvent-related sources, based on long-term observations in Hong Kong, are also inconsistent with those in local emission

inventory (Ou et al., 2015). More measurements in different areas are needed to verify the emission trends of NMHCs in China. Shanghai is one of the fast developing megacities in China. Its population has grown from 22.1 million to 24.2 million from 2009 to 2015, the gross domestic product (GDP) increased from 1.50 to 2.51 trillion RMB and achieved 67 % growth by 2015 (Figure S1). As such, energy consumption increased by 13 % during 2009–2015, and the number of vehicles has increased by around 12 % per year (1.47–2.82 million) (China Statistical Yearbook, NBS), which result in large emissions of NMHCs (Figure S1).

Meanwhile, a series of control measures for NMHCs reductions has been introduced in Shanghai, such as strengthening the standards of vehicles, clean fuels actions, updating the standards for industrial coatings and so on, which are summarized in Table S1. Long-term variations of NMHCs concentrations during 2007–2015 in urban Shanghai have been reported (Gao et al., 2017), but characteristics and sources evolution of NMHCs haven't been discussed.

    In this study, continuous online measurements of NMHCs were conducted at an urban site in Shanghai Academy of

Environmental Science (SAES) from 2009 to 2015. Based on the long term measurements, the objectives of this study were (1) to quantify the long-term variations of NMHCs abundance, speciation and ozone formation potentials (OFP) from 2009 to 2015 considering the impact of meteorology and photochemistry; (2) to assess the source evolutions of NMHCs over time, combining with the social and economic activity data; (3) to validate the speciation of emissions inventory primarily. Accordingly, the implications to validate emission inventory and to control NMHCs emission differentially in future were discussed.

# 2 Material and methods


## 2.1 Monitoring site

    Continuous online measurements of NMHCs were conducted from 2009 to 2015 at Shanghai Academy of Environmental Science (SAES, 31.17 N, 121.43 E, 15 m above ground). Detailed description of the site could be found in our previous studies (Wang et al., 2013; Wang et al., 2016; Wang et al., 2020), to which the reader is referred for a detailed description of the site.

Briefly, this site is a typical urban environment in Shanghai (Liu et al., 2019), located at a five-story building at SAES, and the industrial sources are around 50 km away in the south and southwest area. At a large enough temporal scale, measurements at SAES supersite were representative of the air pollution in Shanghai.

## 2.2 Measurements of NMHCs

    Fifty-three kinds of NMHCs were measured by an online high-performance gas chromatograph with a flame ionization detector

(GC-FID) (Chromato-sud airmoVOC, France) which consists of two separate systems for detection of the C2–C6 hydrocarbons


(GC1) and the C6–C12 (GC2) hydrocarbons, respectively. The sample was preconcentrated through two-stage traps, and then thermally desorbed, followed by a flame ionization detector (FID). The time resolution of the instrument was 30 min. Details about the measurements can be found in Wang et al.(2013) and Wang et al. (2020a). Data quality assurance and control analysis was described comprehensively in Text S1. Briefly, the fifteen times multi-points calibrations with standard gas (Spectra, USA)

were found to be repeatable (Figure S2). The single-point calibration with standard materials (n-Hexane and Benzene) in tube, was performed each day to assure data consistency (Figure S2). As shown in Table S2, responses of most species agreed well with standard gas over studying period. In general, the correlation coefficient ranged from 0.949 to 0.999, and the detection limits ranged from 10 to 470 pptv. The data efficiency was drawn as Figure S3, with an average of 80 % in this study. P value associated with the standard F-statistic test were used to obtain the statistical significance of the interannual trends of NMHCs

concentrations and sources.

The acetylene in 2015 was excluded due to the trap was out of work for concentrating acetylene but still effective for other species in 2015. Several studies also found that the accuracy and uncertainty of the measurement of acetylene was considerably larger than other species through same measuring method (Apel et al., 1994;Bon et al., 2011).

### 2.3 Multiple linear regression model

A stepwise multiple linear regression model (MLR) model was conducted to analysis the effects of meteorology on NMHCs variability. A number of studies have examined meteorological influences on air pollutants (like ozone and fine particles) variability with MLR analysis (Tai et al., 2010;Otero et al., 2018;Zhai et al., 2019;Li et al., 2019c). The model fit the deseasonalized and detrended monthly NMHCs mean time series to the related monthly mean meteorological variables with stepwise regression, by adding or deleting terms based on their independent statistical significance. Then meteorology-driven

anomalies could be excluded, to get to the meteorology-corrected values. To rule out the interference from meteorology on trends of NMHCs, we performed MLR model with ambient NMHCs and meteorological variables (temperature, wind speed and direction, air pressure and relative humidity) during the long-term period.  Detailed processes were described in Supplement (Text S2).

The effect of solar radiation which played important roles in NMHCs photochemistry has also been studied through estimating

the photochemical loss rate of NMHCs over the periods (Text S3, Figure S5).

### 2.4 Positive matrix factorization (PMF) model

The US EPA PMF v5.0 was applied for ambient NMHCs source apportionment analysis. More detailed information about this method can be seen elsewhere (Paatero and Tapper, 1994;Norris et al., 2014). The species with high uncertainty or high reactivity were excluded from the input to reduce the possible bias of the modeling results, except for the source tracers. 16 target species,

namely, Ethane, Ethylene, Propane, Propylene, Isobutane, n-Butane, Isopentane, n-Pentane, Benzene, Toluene, Ethylbenzene, m,p-Xylene, Styrene, o-Xylene, n-Nonane, n-Decane, were chosen for the source apportionment analysis, which contribute >75% of the total NMHCs together.

### 2.5 Activity data

The social and economic activity data during 2009–2015 were mainly gathered and assigned from China Statistical Yearbook

(2009–2015) released by the National Bureau of Statistics (NBS), including the annual urban population, gross domestic production (GDP), energy (natural gas and gasoline) consumption, and numbers of vehicles in Shanghai. Data on the industrial activities (paint production) and petro-chemical output were collected from the Shanghai Statistical Yearbook (2009–2015)



(SMSB). The annual concentrations of pollutants (CO, $NO_2$) were from the Shanghai Ecological and Environmental Bulletin (2009–2015) (SMBEE).

## 3 Results and Discussion

### 3.1 Temporal variations of NMHCs levels

#### 3.1.1 Interannual trends in Shanghai

The average mixing ratio of NMHCs is 25.5±1.9 ppbv from 2009 to 2015 (standard deviation describes the yearly variabilities). Both trends of the observed and meteorology-corrected NMHCs were presented in Figure 1, which were very similar with each other. The annual NMHCs mixing ratios showed no statistical trend from 2009–2015 as well as those corrected by meteorology. It was expected that the meteorology in a specific region was relatively stable on an annual scale. This variation of NMHCs concentrations disagreed with a persistent growth of NMHCs emissions inventory (Li et al., 2019f), suggesting discrepancy between observations and emission databases in Shanghai. Different trends of the major chemical group of NMHCs were also shown in Figure 1, discussed as following below.

The mixing ratios of alkanes increased from 10.5±2.7 ppbv in 2009 to 13.8±3.4 ppbv in 2015 with a statistical observed trend (3.8% $yr^{-1}$) and meteorology-corrected trend (5.5%$^M$ $yr^{-1}$) ($p < 0.05$), respectively. It should be noted that the dominant driver to the growth of alkanes was ethane (13.5% $yr^{-1}$, $p < 0.05$, Table S4), which is a known marker of natural gas production and usage (Derwent et al., 2000;Song et al., 2007). The increasing consumption of natural gas in Shanghai (3.4 to 7.7 billion $m^3$) during 2009–2015 (Shanghai Statistical Yearbook 2009–2015) confirmed the result above. The mixing ratios of alkenes decreased from 4.3±1.1 ppbv (2009) to 3.2±1.0 ppbv (2015), and the average decrease rate was 1.3% $yr^{-1}$ (0.5%$^M$ $yr^{-1}$), without a statistically significant trend ($p > 0.05$).

The annual aromatics mixing ratios decreased gradually, from 8.4±2.7 ppbv in 2009 to 5.4±1.3 ppbv in 2015 (-6.0% $yr^{-1}$, -4.6%$^M$ $yr^{-1}$, $p < 0.05$), which is close to results found by Gao et al. (2017). Similar decreasing trends of aromatics were also found in other urban areas in China (Ou et al., 2015;Wang et al., 2015;Wang et al., 2017b). As the government has put tremendous effort in controlling emissions from transportation, residential sector and industry (Table S1), the decreasing trend suggested the effectiveness of these measures, especially strengthening emission standards on vehicles and coatings. Acetylene is often considered as a product of incomplete combustion (Liu et al., 2008b). Ambient annual acetylene mixing ratio also exhibited a significant ($p < 0.05$) negative trend, from 3.0±1.0 ppbv in 2009 to 1.7±0.6 ppbv in 2014, with relative rate of -7.8% $yr^{-1}$ (-6.4%$^M$ $yr^{-1}$, $p < 0.05$). Declines of both aromatics and acetylene in these years could mostly attribute to emission controls in transportation, i.e. elimination of old vehicles and strengthening the fuel and vehicle emission standards, as well as the clean fuel action in other sectors (Table S1).

The trends of major NMHC species with high abundance and reactivity were further analyzed (Table S4) by their measurements directly, considering the meteorology-corrected trends agreed well with observed trends mentioned above. Statistical decreases ($p < 0.05$) were found for benzene and toluene concentrations during 2009–2015, with the annual rate of -6.0% $yr^{-1}$ and -8.3% $yr^{-1}$, while the ethane increased at the rate of 13.5% $yr^{-1}$ significantly. For the other alkanes with large contributions to concentration, like propane, butanes, and pentanes, no statistic trends were observed even though there were some increases. Similarly, some decreases were observed for alkenes like ethylene and propylene and C8 aromatics concentrations but without statistic trends.

The photochemistry effect on NMHCs trends was also tested in this study (Text S3), combined estimation and observation of oxidants concentrations. The insignificant trends of photochemical loss rate suggested little effect of oxidation capacity change on evident observed NMHCs trends (Figure S5). This was again supported by comparison with the trends based on nighttime





measurements (Table S5). It suggested the increasing trend of atmospheric oxidation capacity indicated by $O_3$ played limited roles in the variations of NMHCs in the present study. Thus, the observed results were discussed directly below.

### 3.1.2 Interannual trends in different regions of China

To better understand the interannual variations of NMHCs in Shanghai, we overviewed other studies in the Beijing-Tianjin-
Hebei (BTH) (Song et al., 2007;Duan et al., 2008;Xie et al., 2008;Wang et al., 2010;Liu et al., 2016;Li et al., 2015b), the Yangtze River Delta (YRD) (Geng et al., 2010;Cai et al., 2010;Wang et al., 2013;An et al., 2014;Xia et al., 2014;An et al., 2017;Mo et al., 2017) and the Pearl River Delta (PRD) (Li and Wang, 2012;Zou et al., 2015;Huang et al., 2015), to explore the trend of NMHCs in eastern China as shown in Figure 2. 16 studies were included in Figure 2, which met the following constraints: (i) measurements located in urban or suburban areas, similar to our study. The NMHCs in three main populated areas were studied
most, with increased concern; (ii) measurements covering at least two seasons in each year over the period 2005–2015, to avoid seasonal influences. Long-term measurements in Beijing in summer during 2002–2013 were also mentioned in the present study, which investigated comprehensively the trends of NMHCs and their sources (Wang et al., 2015;Zhang et al., 2014); (iii) reports measuring NMHC species mainly were prior to choose, close to which in our study. The contribution of species which were same to those in our study was summarized (79%−100%) (Table S6), indicating little bias with different species measured in selected
studies. In addition, three long-term trends of NMHCs reported in these regions were also included (Gao et al., 2017;Zhang et al., 2014;Ou et al., 2015;Wang et al., 2017b). Detailed information about these studies was presented in Table S6.

With above limitations, it aimed to explore the trends of NMHCs in urban areas in China, rather than making comparisons among individual observations in this study. As shown in Figure 2, there was a declining trend in Beijing during summer from 2005 to 2013, while it showed a growth in inventory (Zhang et al., 2014;Wang et al., 2015). Based on the previous result of
Shanghai (Gao et al., 2017), NMHCs decreased by 0.44 ppbv yr$^{-1}$, unfortunately without statistical significance, which was difficult to compare with this study. No obvious changes of NMHCs over these years were found in PRD region over the period. Concluded from measurements above, there was no evident increase of NMHCs in China from 2005 to 2015, different from the increasing trend of emissions estimated by Li et al. (2019f). Considering the fast development through over all social sectors in eastern China in the past years, the relative flat trends of NMHCs indicated the effectiveness of control actions implemented, and
also revealed the need of more work on NMHCs emission inventory in China.

### 3.2 Characteristics of NMHCs

### 3.2.1 NMHCs Compositions

The annual mixing ratios of NMHCs and compositions were presented in Figure 3. Over past seven years, the average composition of NMHCs was mainly characterized by alkanes (mean proportion of 53%), aromatics (27%), followed by alkenes
(12%), and to a lesser extent by acetylene (8%). While interannual NMHCs mixing ratios were similar in these years, their compositions showed some changes (Figure 3), with reduced proportions of aromatics (from 29% in 2009 to 21% in 2015, the same below), increased share of alkanes (from 46 % to 57 %), and relative stable contribution of alkenes (from 15% to 13%) and acetylene (from 10% to 8%).

NMHCs compositions in this study were compared with previous studies conducted in Shanghai and other urban areas in China,
as shown in Figure 3. The alkanes and alkenes accounted for about 53 % and 12 % in Shanghai, slightly lower than those in other areas (54–64 % and 14–23 %, respectively). While, a higher proportion of the aromatics contributed to NMHCs in Shanghai (27 % on average) than in other areas (9.4–24 %). As reported, aromatics could account for ~50%−70% of total NMHCs in source profiles of solvent use (Wang et al., 2014). Besides, more than 60% of aromatic were reported from solvent usage and industrial





productions based on source apportionment studies in Shanghai (Geng et al., 2010). In the meantime, solvent use has gained a
considerable amount of attention due to its contribution to NMHC emissions in Shanghai, accounting for 25% in 2007 according
to the emissions inventory (Huang et al., 2011), ~32% of the measured ambient NMHCs during 2007–2010 (Cai et al., 2010).
Thus, to some extent, the larger contribution of aromatics may indicated larger roles of industry/solvent use in Shanghai than
those in other areas of China.

### 3.2.2 Ozone formation potentials of NMHCs

To study the contribution of grouped hydrocarbons to ozone pollution, the ozone formation potentials (OFP) of NMHCs were
calculated combining with the constant of maximum incremental reactivity (MIR) as reported (Carter, 1994;Zheng et al., 2018b).
Figure 4 shows that the overall OFP dropped by about 25 %, from 143 ppbv in 2009 to 107 ppbv in 2015, at a rate of 3.8% yr$^{-1}$($p<0.05$). Aromatics contributed to 55–66 % of the OFP over the past years, far more than other groups (alkanes, 14–18%;
alkenes, 20–26%) as shown, indicating the aromatics always played a dominant role in ozone pollution in Shanghai even though
it showed a downward trend.

The average MIR (sign as $MIR_{avg}$) of NMHCs was calculated as the ratio of the total OFP to the total mixing ratio of NMHCs
(Wang et al., 2013;Wang et al., 2014). As shown in Figure 4, the $MIR_{avg}$, significantly decreased by 0.14 ppbv $O_3$/ ppbv NMHCs
(-2.3% yr$^{-1}$, $p<0.05$) from 2009 to 2015, to some extent suggesting the effectiveness of NMHCs controlling measures. This
decrease was mainly attributed to the decreased contribution of aromatics as mentioned above (Figure 3), which had relatively
high MIR.

Both reduction of concentration and OFP of NMHCs should be beneficial for mitigation of $O_3$ pollution. However, observed $O_3$
concentration has been increasing as reported (Gao et al., 2017;Xu et al., 2019). This was mainly because $O_3$ production
Shanghai was under NMHCs-limited regime (Cai et al., 2010;Gao et al., 2017), and limited reduction of NMHCs compared to
NOx reduction from 2009 to 2015 played important roles in $O_3$ increase (Xu et al., 2019). It could be expected if without the
decrease of NMHCs and related $MIR_{avg}$, the $O_3$ pollution could be worse than observed at present. Further reduction in NMHCs
and cooperative control of the precursors will be critical to promote the mitigation of $O_3$ pollution in Shanghai in future.

### 3.3 Typical species ratios

### 3.3.1 Ratio of toluene to benzene (T/B)

Ratios between species, such as toluene to benzene (T/B), have been commonly used to make simple identification and
judgements on NMHCs sources. Toluene is usually associated with solvent use in the painting, coating and printing processes in
addition to vehicle exhaust. However benzene has been prohibited in industry solvents or consumer products as its
carcinogenicity, which mainly come from automobile exhaust in urban areas. Thus, higher T/B ratio in urban areas typically
suggest that more emission from industrial activities. Based on tunnel and roadside research, the T/B ratio varies within the range
of 1–2 when main source is vehicle exhaust (Barletta et al., 2005;Barletta et al., 2008;Zhang et al., 2012;Song et al., 2018;Li et
al., 2019a), and T/B >2 indicates that there are important emissions from paint and solvents, besides traffic source.

The annual mean T/B ratios were calculated by the nighttime (0:00–4:00 LT) measurements to exclude the interference of
photochemistry. As shown in Figure 5, the mean T/B ranged from 2.7 to 3.8 (3.4±0.4), with a statistical decreasing trend by 0.15
yr$^{-1}$ after 2010. It suggested the decreasing contribution of solvent usage after 2010 due to the promulgation of stricter standards
of industrial painting sector in Shanghai (as shown in Table S1). The increase of T/B ratio from 2009 to 2010 might be due to
more activities related to solvent usage for the holding of the 2010 World Expo in Shanghai, before which nearly all of the
external walls of the building in Shanghai were a new paint. Overall, the ratio of T/B in Shanghai was similar to the values of





Guangdong in 2014 (Song et al., 2019) and Hong Kong in 2011 (Huang et al., 2015), indicating that industry-related emissions were important sources of NMHCs in these areas. The relatively lower ratios reported in urban Beijing suggest the main source is vehicular emissions (Zhang et al., 2014;Liu et al., 2017).

### 3.3.2 Emission ratio of NMHCs/acetylene

The emission ratio (ER) is the ratio between two species in their emissions, and generally an inert tracer (eg., CO, acetylene) can be selected as the reference species (Warneke et al., 2007;Borbon et al., 2013), which could be approximately estimated by the measurements in the ambient. Here we derived emission ratios of NMHCs versus acetylene from slopes of orthogonal distance regression (ODR) fits (Wu and Yu, 2018) on the measurements during nighttime (0:00–4:00 LT), in order to exclude the effects of photochemistry (Warneke et al., 2007;Bon et al., 2011;Borbon et al., 2013;Li et al., 2019e;Yuan et al., 2012).The slope determined by fitting two species actually represented the ratio of their concentration changes resulting from the emissions rather than the physical mixing processes.

For this method of determining the emission ratios, assumptions as followed must be valided: (1) the emission of measured NMHCs was proportional to the emission of acetylene. Acetylene has often been used as a marker for urban emissions (Boynard et al., 2014;Chen et al., 2016;Warneke et al., 2007). Most species also showed significant correlations with measured acetylene, with the averaged correlation coefficient (R) ranged from 0.5 to 0.8; (2) as a relatively inert tracer, acetylene emissions estimated in emission inventory assumed to have lower uncertainty than others.

Then the ER of individual species to acetylene obtained by the measurements in this study were compared with those based on the Multi-resolution Emission Inventory for China (MEIC) (Li et al., 2019f;Zheng et al., 2018a), as shown in Figure 6 (Table S7). Here, we obtained emission data related to our study from MEIC, that is, NMHCs emissions from 2009 to 2014 in Shanghai, due to lack of acetylene in 2015 (Section 2.2). For most NMHCs, the determined emission ratios based on measurements were approximately a factor of 2 lower than those in emission inventory. There was an underestimation of C2–C3 alkanes in the emission inventory, probably because of the underestimation of natural gas usage (Borbon et al., 2013). This finding was consistent with the study by Li et al. (2019b), which compared the NMHCs annual emissions derived from the ambient measurements and emission inventory in Beijing. These discrepancies revealed a need to validate the speciation of the current emission inventories with more evidence at large spatial or temporal scales, and then evaluation of impact of NMHCs on air quality would be improved.

### 3.4 Source apportionment of NMHCs

### 3.4.1 Source identification

The PMF model was performed based on the samples collected during 2009–2015. Five factors were determined according to the source profiles (Figure S6, and monthly variations of these factors were analyzed. Factor 1 largely consisted of toluene, ethylbenzene, and xylenes, which were reported as major constituents of solvent usage (Wang et al., 2014;Yuan et al., 2010). Relatively good relationships between these compounds (r > 0.8, Figure S7) also indicate that they probably come from similar sources. This source exhibited a little month-to-month variation, with a minimum in February, which may be influenced by reduce of industrial activities during Spring Festival holidays in China. Thus factor 1 was interpreted as solvent usage profile.

Factor 2 was characterized by abundant ethane and propane, which are commonly come from use of natural gas (NG) (Sun et al., 2016a;Li et al., 2015a;Zheng et al., 2018b). Furthermore, factor 2 also gave significant contribution to benzene which has relatively long lifetime comparing with those of ethane and propane. It means factor 2 is probably one kind of relatively aged source. The monthly variation of factor 2 present higher concentrations in cool seasons than those in warm seasons, which was



different from the flat pattern of the consumption of NG (Figure S8). Considering the prevailing wind in winter of Shanghai is from the northwest where experienced heavy air pollution, the regional transport might play important roles in NMHCs in winter of Shanghai (Wang et al., 2020b). Therefore, it was referred to as mixed source of NG and regional transportation. Measurements of more indicative species were essential to apportion these two kinds of sources in future.

Factor 3 explained most of the C3–C4 alkanes, which are commonly associated with liquefied petroleum gas (LPG), diesel and

gasoline exhaust (Wang et al., 2017a;Na et al., 2004). In urban Shanghai, LPG usage was around 2.9%±0.5% in annual resident energy consumption per capita according to Shanghai Statistical Yearbook (2009–2015), then it could make little contribution to NMHCs emissions. In addition, as shown in Figure S6, factor 3 also contained around 26% of toluene, and 12% of ethylene, which were considered to be important components of vehicle exhaust and support the identification of this factor. Thus, factor 3 was considered as vehicle exhaust.

Factor 4 was associated with high percentages of ethylene and benzene, which have been identified in the emissions of the petrochemical industry (Song et al., 2007;Liu et al., 2008a;Song et al., 2018). Meanwhile, there were propene, toluene and styrene, which were reported to be the major material/products of the petrochemical source (Mo et al., 2017). Therefore, this factor was referred to as petrochemical industry.

Factor 5 was characterized by high levels of isopentane and n-pentane, which are common tracers of gasoline evaporation

(Barletta et al., 2005;Morikawa et al., 1998). And around 30% of n-decane contributed in this factor, and a high level of n-decane is a good marker for diesel emission (Shen et al., 2018;Watson et al., 2001). It's noted that higher emissions of this source in summer than other sources due to higher temperature (Figure S6). So it was identified as fuel evaporation.

### 3.4.2 Source contributions

The annual and monthly source contributions were presented in Figure 7, as well as the previous results by both observations and

inventories in Shanghai.

As shown in Figure 7, vehicular emissions were a dominant source (20–30%) in Shanghai during 2009–2015, which gradually decreased from 30% in 2009 to 24% in 2015. Contributions obtained in this study in 2009 and 2010 were consistent with the results of Cai et al. (2010) (25%) and with the values of inventory in 2010 (23%) (Huang et al., 2011;Wang et al., 2013). Vehicular emissions contribution from PMF both in this study and the previous studies were higher than the value of 13 % in

inventory of 2013 reported by Wu and Xie (2017) and 6% in MEIC (averaged proportion of 2009–2015) (Zheng et al., 2018a;Li et al., 2019f). Besides, the contribution of petrochemical industry (15–23%) was fairly similar to that in reports by Cai et al. (2010) (28%) and MEIC (19%), but much lower than those of Huang et al. (2011) and Wu and Xie (2017).

The solvent usage was another dominant source of NMHCs with a proportion of 21–32%, which was comparable to contributions in inventories (27% in 2010, 31% in 2013), but lower than what in MEIC (46%). It seemed a possible

overestimation on emissions from solvent use in the MEIC inventory, similar to the findings in Beijing (Wang et al., 2015).

Fuel evaporation (14–20% in this study) source contributions in NMHCs exhibited large differences among all the results listed in Figure 7. A unified identification and classification system of emission sources would be helpful to make comparisons. The contribution of NG and regional transport contributed around 9–21 % to NMHCs with an increasing trend. Moreover, monthly distributions of these sources clearly showed that fuel evaporation has a higher contribution in summer, while NG and regional

source was higher in winter.

### 3.4.3 Interannul trends of sources

Figure 8 shows the temporal changes in NMHCs concentrations contributed by PMF-resolved factors and related activities in





Shanghai during 2009–2015. The NMHCs concentrations from vehicle exhaust decreased significantly during these seven years, at an annual rate of 6.5% $yr^{-1}$, which displayed fair correlations with nitrogen dioxide ($NO_2$) and carbon monoxide (CO), as

markers of vehicle exhaust in urban (Shanghai Ecological and Environmental Bulletin (2009–2015)) (SMBEE). The concentrations from vehicle exhaust decreased dramatically by 46% from 2009 to 2012, but showed no obvious change after 2012, which was similar to the trends of elemental carbon in Shanghai during 2010–2014 (Chang et al., 2017). This probably suggest that with the elimination of old vehicles and the improvement of fuel quality, and the follow-up control efforts may not be enough to offset the increase in emissions caused by the increase in vehicle fleets. Therefore, it's still a challenge to reduce the

emissions from vehicle exhaust in the future.

    Solvent usage contributions decreased by -8.0% $yr^{-1}$ (p<0.05) statistically after 2010, which indicated the effectiveness of efforts, like the enactment of local emission standards for the above source (Table S1), have been made in the past years. The increase from 2009 to 2010 might be due to the more paint usage for the 2010 World Expo holding in Shanghai. The trend of solvent usage contribution was inconsistent with the increase in paint production in Shanghai (16% $yr^{-1}$) (Shanghai Statistical Yearbook

(2009–2015)) (SMSB). Besides, during 2009–2015, the production of architectural coatings in Shanghai increased from 0.46 to 1.03 million ton, and the production of automobiles increased from 1.25 to 2.42 million vehicles. This partly was expected as the emissions related to the usage, such as printing, furniture manufacturing and automobile coatings not to the production. Meanwhile, it indicated the possible growing trend in emissions, due to growth in above activities, was effectively offset by measures implemented in solvent usage.

The contribution of petrochemical industry decreased by 9.4% $yr^{-1}$ with statistical significance. But it was much higher in 2009 than that in later years, probably affected by new urban infrastructures. According to Shanghai Statistical Yearbooks, investment in urban infrastructures in 2009 was around two times high than that in 2011. Noted that no obvious change was found between 2011 and 2015 (p=0.14), consistent with value of petro- and fine- chemistry output (SMSB), which suggested the related control measures may be not stringent enough to reverse the growing trend in emissions from petrochemical industry.

The NMHCs concentrations contributed by natural gas usage, mixing with some regional transportation, increased by 9.2% $yr^{-1}$, and agreed with the increasing consumption of natural gas (from 3.4 to 7.7 billion $m^3$ , 15% $yr^{-1}$) during 2009–2015 (China Statistical Yearbook (2009–2015)) (NBS). It also showed a slight growth in the contributions of fuel evaporation to NMHCs concentrations over the past years, indicating the absence of effective control for evaporative sources.

    Overall, vehicular emissions and solvent usage were two largest contributors of NMHCs in Shanghai, showing a declining trend

during 2009–2015, which suggested the effectiveness of controlling measures vehicles and solvent usage. The petrochemical industry emissions also decreased significantly, while after 2011, there were no significant decline of petrochemical industries, which needed more efforts to reduce their emissions. Especially, the increase contributions of natural gas usage and fuel evaporation should be paid more attention to due to the increasing consumption of natural gas and gasoline.

### 4 Conclusions

Fast developing over all the social sectors and toughest-ever clean air policy implementation occurred over the past years in the nationwide of China. A full review of the air pollutants concentrations response to the policy at large temporal scales was beneficial to design future clean air policy. In this study, long-term measurements from 2009 to 2015 of NMHCs were employed to investigate the evolution of NMHCs characteristics and their sources, as well their response to the activity data in Shanghai.

    As a result, the mixing ratios of total NMHCs and four chemical groups averaged as 25.8±2.3 ppbv, 13.6±1.7 ppbv (alkanes),

3.2±0.6 ppbv (alkenes), 6.9±1.2 ppbv (aromatics) and 2.1±0.4 ppbv (acetylene), respectively, from 2009 to 2015. The long-term



trend of NMHCs showed no statistics changes over the period, similar to previous observations in other areas in China, which was inconsistent with the increasing trend indicated by emission inventories. Different trends of the major chemical groups were observed. Specifically, alkanes, aromatics and acetylene increased/decreased by +3.8 % yr$^{-1}$, -6.0 % and -7.8 % yr$^{-1}$ ($p<0.05$), respectively, whereas alkenes showed no statistic interannual trend through the seven years. The meteorology and photochemistry which might impact the NMHCs measurements were tested as negligible effects on an annual scale. OFP showed a significant decline with 3.8% yr$^{-1}$ ($p<0.05$) from 2009 to 2015, which was beneficial to ozone pollution in Shanghai in NMHCs-control regime.

NMHCs source apportionment based on PMF model showed that five sources dominant the NMHCs emissions in Shanghai, including vehicle exhaust, solvent usage, NG mixing some regional transport, petrochemical industry and fuel evaporation, contributed 20–30%, 22–32%, 9–21%, 15–23%, 14–20% (range of interannual proportion) over the period of 2009–2015, respectively. Decline of NMHCs concentration from vehicle exhaust (-6.5% yr$^{-1}$, $p<0.05$) indicated the effectiveness of control measures taken in vehicular emissions, even with a growing vehicles population. Solvent usage contribution to NMHCs concentration decreased by -8.0% yr$^{-1}$ ($p<0.05$) statistically after 2010, which indicated the effectiveness of efforts, like the enactment of local emission standards for the above sources, have been made in the past years. Petrochemical emissions after 2010 showed no significant decline, which needed more reductions. The distinct growth of natural gas source (mixing with some regional transportation) (9.2% yr$^{-1}$, $p<0.05$) was mainly originated from increasing consumption (15% yr$^{-1}$) of natural gas probably due to the clean fuel plans in industrial boilers and increasing consumption of residential sector. Fuel evaporation showed no evident change, suggesting more efforts to reduce their emissions in the future. Overall, vehicles and solvent use (except 2009) presented a decreasing trend over the period; however, petrochemical industrial emissions had no positive trend after 2011; and emissions of natural gas and fuel evaporation became more and more important in Shanghai.

Our study confirmed a positive effect of the efforts that have been placed into controlling measures in vehicle exhaust and industry-related sources including solvent usage, which were mainly centralized emissions easily controlled through strengthen standards. While, it seemed that more effective measures for petrochemical industries, generally fugitive emissions, were essential in future. The increase contributions of natural gas usage and fuel evaporation should be paid more attention to as the quick increase of natural gas and fuel consumption due to the policy of energy structure reformation in China. Meanwhile, it deserved more attention on further reductions of NMHCs and cooperative control with NOx, which were critical to mitigate O$_3$ pollution. Our results provided some new insights into future clean air policies in megacities of China. Additionally, it provided further evidence that the long-term trend of atmospheric NMHCs was inconsistent with the increasing trend indicated by emission inventories, indicating that the emission inventories are needed to be further improved for more accurately evaluating the impact of NMHCs on regional air quality.

*Data availability.* The data can be accessed upon request to the corresponding author.

*Author Contribution.* HW conceived and designed the study and wrote the paper. QW and SJ contributed to the online measurements and helped in its discussion. JA, YG, CH, RY, and HD assisted with data collection. QZ and ML provided emission inventory data in Shanghai. LL, SL, ST, QH, JL and CC helped with scientific interpretation and discussion. HW and TC revised and polished the article. YP performed the data analysis and prepared the manuscript with contributions from all co-authors.

*Competing interests.* The authors declare that they have no conflict of interest.

*Acknowledgments.* This research has been supported by the National Key Research and Development Program of China (No. 2018YFC0209800, 2017YFC1501405), the Shanghai Science and Technology Commission of the Shanghai Municipality (No. 18QA1403600, 20ZR1447800), the National Natural Science Foundation of China (No. 41775129).



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

**Figure 1.** The trends of monthly mixing ratios of NMHCs and the major chemical categories (ppbv) from observation (in black) and meteorology-corrected data (in red) during 2009–2015. Average annual change rate (ppbv/% yr$^{-1}$) was derived from yearly values.






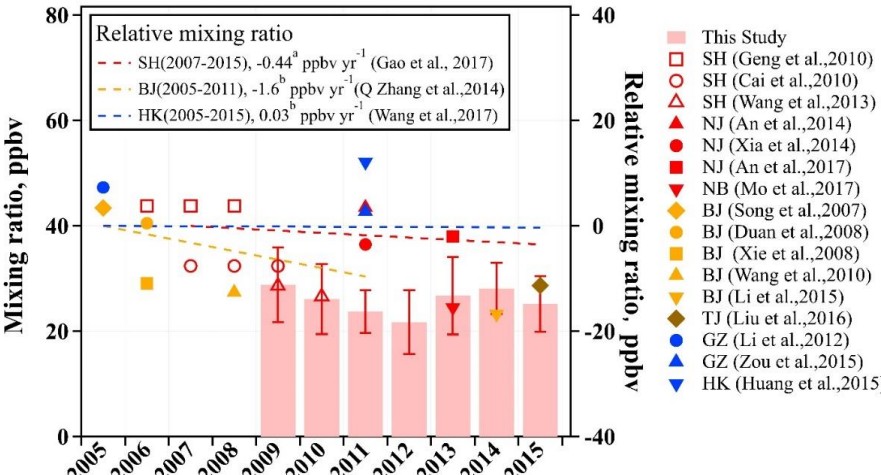

**Figure 2.** Comparison of trends of NMHCs in eastern China during 2005–2015 (SH: Shanghai; NJ: Nanjing; NB: Ningbo; BJ: Beijing; TJ: Tianjin; GZ: Guangzhou; HK: Hong Kong); Bars represent results in this study (left axis); Markers mean average mixing ratios of NMHCs derived from other studies (left axis); Dashed lines mean slopes derived from previous studies presented here by relative mixing ratio to first year (the value of first year set as 0)(right axis), [a] not giving statistics data, [b]

without statistical significance; Error bars represent one standard deviation of monthly averages.





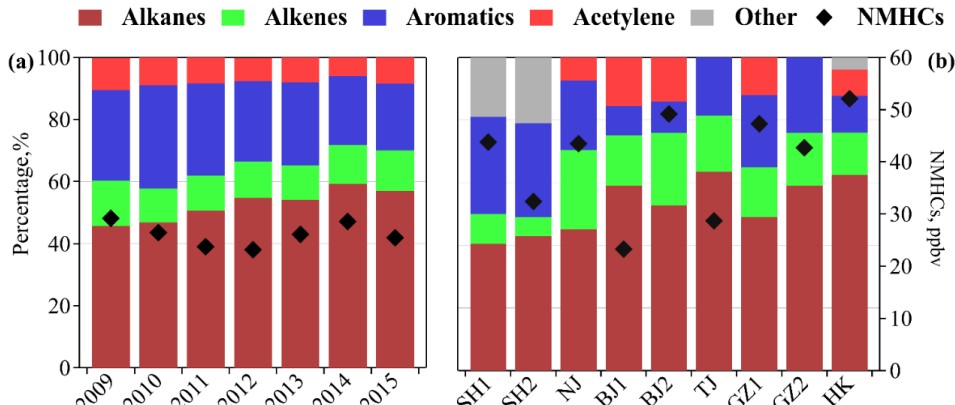

**Figure 3.** Compositions and levels of NMHCs in this study (a) and previous studies in other areas in China (b). SH1:2006–2008 in Shanghai's urban (Geng et al., 2010); SH2:2007–2010 in Shanghai's urban (Cai et al., 2010b); NJ:2011 in Nanjing's urban (An et al., 2014); BJ1:2014 summer in Beijing's urban (Li et al., 2015); BJ2:2015 winter in Beijing's urban (Liu et al., 2017); TJ:2015 in Tianjin's urban (Liu et al., 2016); GZ1:2005 in Guangzhou's urban (Li and Wang, 2012); GZ2:2011 in Guangzhou's suburban (Zou et al., 2015); HK:2011 in Hong Kong's roadside (Huang et al., 2015).





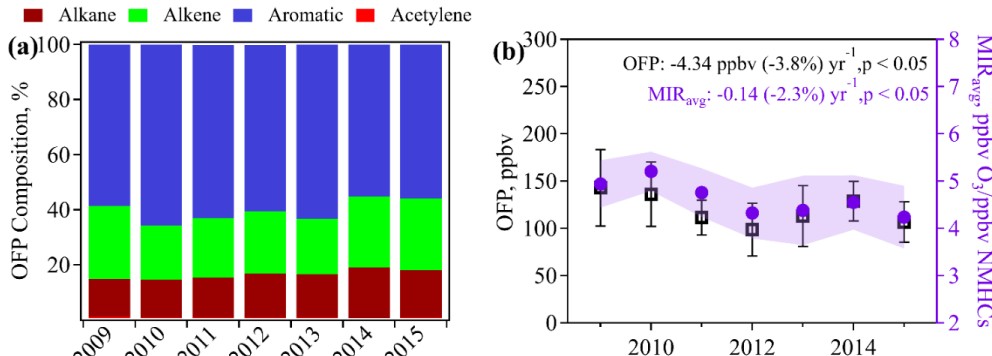

**Figure 4.** Interannual variations of (a) contributions of NMHCs to OFP; (b) OFP of NMHCs and MIR$_{avg}$ during 2009–2015 (Error bar and shaded area represent one standard deviation from monthly OFP and MIR$_{avg}$, respectively).





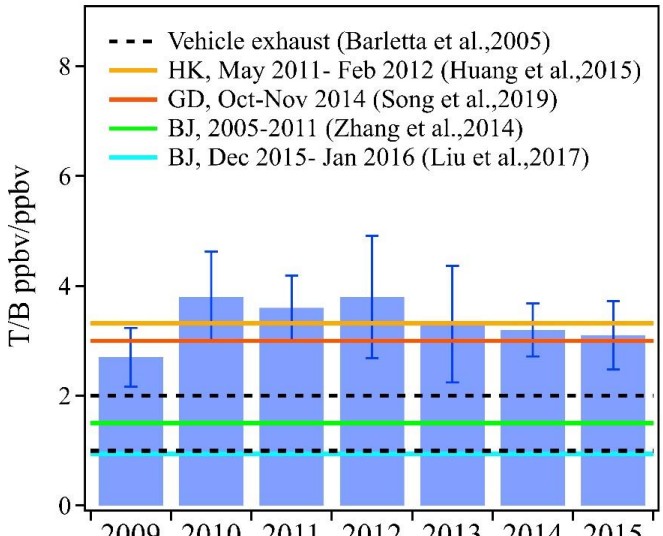

**Figure 5.** Interannual variations of T/B ratio from 2009 to 2015, B: Benzene; T: Toluene. The dashed lines mean the typical ratio of vehicle exhaust (1–2), results from roadside samples in 25 Chinese cities in 2001 (Barletta et al., 2005). Solid lines mean

values from ambient data measured at Beijing (BJ), Guangdong (GD) and Hong Kong (HK). Error bars mean one standard deviation from monthly averages.





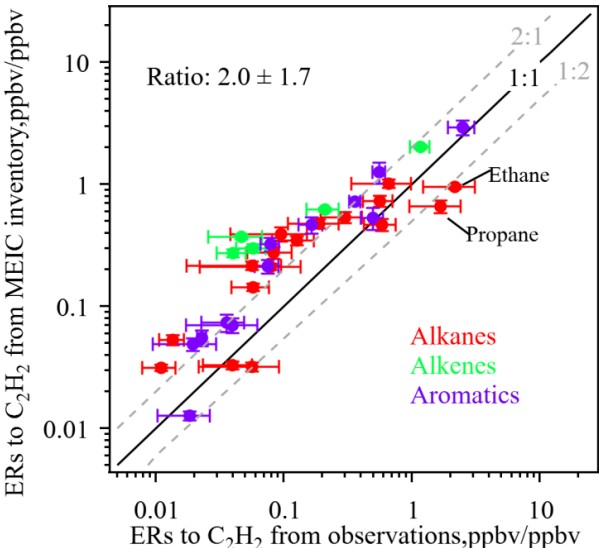

**Figure 6.** Comparison of the emission ratios of NMHCs to acetylene based on measurements in this study with those from
emission inventory data. Each data point means one compound. The black solid line indicates the 1:1 relationship. The grey
dashed lines mean 1:2 and 2:1, respectively. Error bars associated with these ratios represent year-to-year variability.


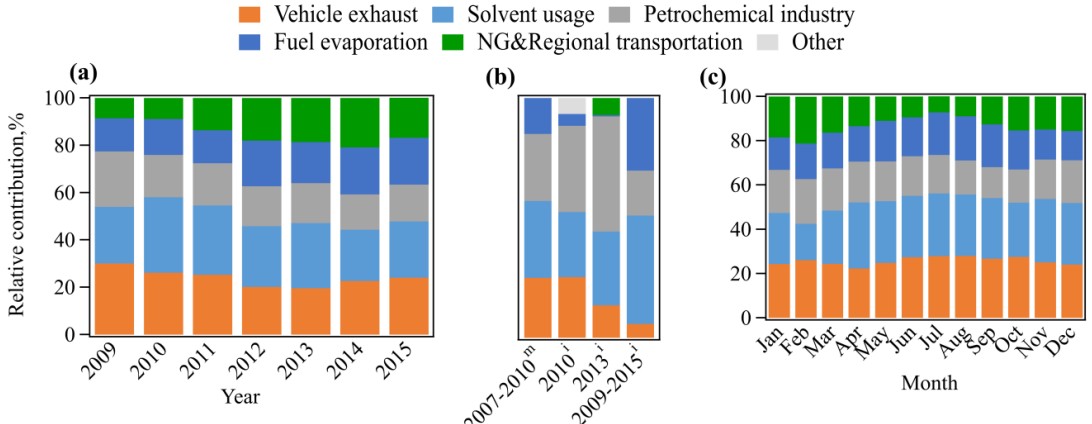

**Figure 7.** Source contributions of NMHCs. (a and c): annual and monthly variations in this study; (b): results from other studies.
[m] measurements (2007-2010: measurements during 2007–2010 by Cai et al.,(2010b)); [i] inventory (2010: inventory in 2010 estimated by Huang et al.,(2011), 2013: inventory in 2013 described in Wu et al.,(2017), 2009–2015: averaged result of 2009–2015 from Multi-resolution Emission Inventory (Li et al., 2019f;Zheng et al., 2018a).

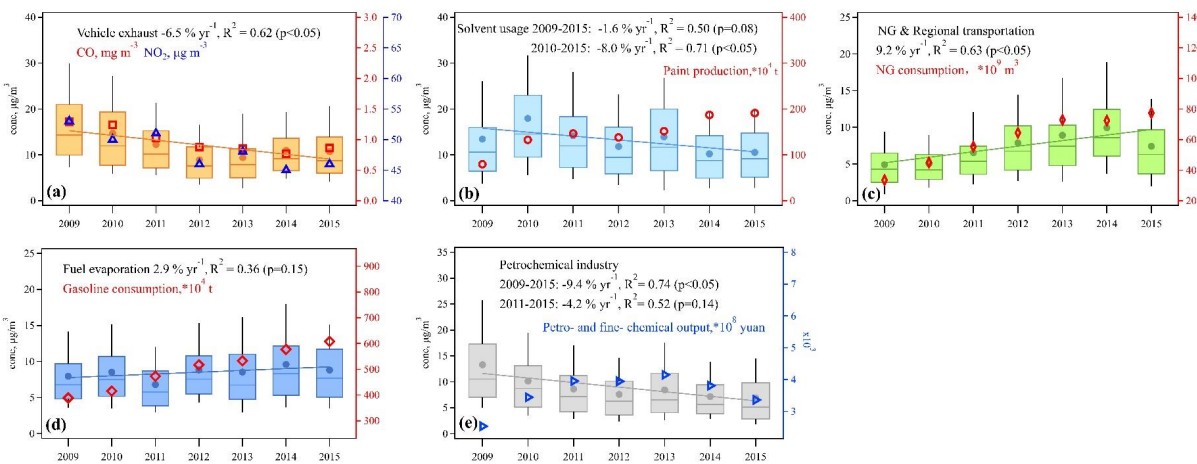

**Figure 8.** Trends of NMHCs concentrations (conc) contributed by PMF-resolved sources (a–e) and related activities data in Shanghai. Each box represents the 25–75th percentiles. The line across the box represents the median value. The whiskers that extend from the bottom and the top of the box represent the 10–90th percentiles. Dots in the box mean annual averages. Markers represent related activities data. (CO and $NO_2$ concentration from Shanghai Ecological and Environmental Bulletin; paint production and petrochemical output from Shanghai Statistical Yearbook; NG and gasoline consumption from China Statistical Yearbook).