# Peer review of "Long-term measurements (2009–2015) of non-methane hydrocarbons (NMHCs) in a megacity of China: implication for emission validation and source control"

_Atmospheric Chemistry and Physics, 2020_

## Referee Comment (RC1) · Anonymous Referee #2 · 8 Jan 2021

The authors present an analysis of long-term VOC measurements in Shanghai. The main results are that ambient mixing ratios of alkanes showed an increasing trend, while aromatics and acetylene decreased. Alkenes did not reveal a trend. According to a PMF analysis emissions from vehicles, petrochemical industry and solvent use decreased, while emissions of natural gas and evaporation increased. Those results appear to be non-consistent with the emission inventory. In terms of O3 formation the authors state that decreasing VOC concentration would yield lower O3 forming potential, even more important for Shanghai as O3 formation would be in a VOC limited

regime in that area.

When I first saw the paper I found it interesting, but when I read the supporting text materials my concerns grew and I had the impression that the conclusions the authors were drawing were not always well justified. I provide my detailed comments below. Apart from that I have been frustrated at times to see citations in the text without listing appropriate references, which is just bad. The use of English language in this paper is often poor and sometimes difficult to decode. It needs polishing.

Page 2, L28: "... of which composition varied significantly". I only saw that ethane and alkenes changed significantly (see Fig. 1 and associated discussion).

Page 2, L28-30: This statement cannot be done as long as biogenic VOCs, foremost isoprene are not considered.

Page 3 L72-77: At some point the authors say the measurement site was 15 m above the ground, then at some point they say the site was at a five-story building. I somehow assume the site was on the rooftop of that building. However, I am not so much interested in how many stories a building would have, but I would rather be interested in what height the building would have and - assuming the site was on the rooftop - what the height of the sampling inlet above the rooftop was, and whether the rooftop had any outlet ducts. I neither saw the citation of Wang et al., 2016 nor Liu et al., 2019 in the list of references. What Wang et al., 2020 paper are the authors referring to, as in the list of references there is Wang et al., 2020a and Wang et al., 2020b....? The last sentence of this paragraph is very generic and vague: what do the authors consider "at a large enough temporal scale"?

Page 3 L79: There have been many auto-GCs in use in the world for decades. Why is this GC in particular a "high-performance" instrument?

Page 4, L81-82: Still, I want to know briefly some standard information about the specifications of the traps, their temperature during preconcentration, the duration of preconcentration, the GC columns and the carrier gas used. Also, what sampling line (material, diameter, length) was used? Chromatograms for each channel should be shown.

Page 4 L83-84: Here the authors refer to Text S1. So I add my comments about Text S1 here as well. What specific VOCs were in the calibration gas? At what concentration were the single-point calibrations done? Apart from that it is not correct that using single-point calibration ensures the stability of the instrument. I do not understand the last sentence in the first paragraph ("Then...later years"). The authors state "...concentrations of most species measured agreed well with standard gas". How do the authors know this when actually the standard gas is used for calibration? The uncertainty should be calculated for each VOC individually and not for a group of VOCs.

Page 4, L86-87: I cannot see in Table 2 that the responses of most species agree well with the standard. In fact many VOCs show slopes well below 1, in particular alkenes, foremost 1,3 butadiene, which is well-known for potential losses in GC preconcentration.

Page 4, L88: Data cannot be "efficient".

Page 4, L91-93: How can a trap be "out of work" for acetylene, while it is still properly working for all the remaining VOCs for an entire year? While it is true that acetylene shows challenges in GC analyses, I find it very astonishing/inconsistent that according to table S2 acetylene shows pretty good values for precision and accuracy. At this point I cannot follow at all the authors' statement in L91-93.

Page 4, L95-105: Here the authors refer to Text S2 and Text S3. So I add my comments about Text S2 and Text S3 here as well. Text S2: I understand that the VOC data in the analysis of this paper has been corrected for meteorology. So why then can the authors make statements about evaporation processes (which depends on temperature) and regional transport (which would depend on wind) later on in the text, e.g. as shown in Figure S6 and other places in the paper?

Text S3: Equation 3-2 was derived for a rural area and in their paper Ehhalt and Rohrer (2000) point out that the coefficients should only be used for that specific rural airmass composition. Why do the authors think that equation 3-2 would be applicable the same way for the highly polluted area of the megacity Shanghai? I do not agree with the authors that Figure S5 shows insignificant trends of chemical loss for NMHCs over seven years. Just looking into L-Alkanes I would assume an overall increase by an order of 30% or so, while L-Alkenes show an overall decrease by about 25%. Also, L-Aromatics and L-Acetylene show high interannual variability, albeit no visible longterm trend. Apart from that I find it quite surprising to see an overall positive trend in OH concentrations (I would guess on the order of 30-50%), which the authors do not comment on at all.

The sentence "While the oxidation of NO3...neglected in this study" is irrelevant as this study did not include biogenic VOCs, anyway.

The comparison analysis in the last paragraph does not mean anything as the "whole-day" data would obviously include nighttime data as well.

Page 4, L108-112: I do not understand a couple of things here: In the first place, I do not see an appropriate definition of uncertainty for individual VOCs here. Anyway, assuming accuracy as shown in Table S2 as a proxy for uncertainty I do not understand why the authors kept VOCs with appreciable uncertainty for the PMF analyses (e.g. ethane, i-butane, n-butane, n-pentane, m/p-xylene) while they stated they wanted to remove VOCs with high uncertainty. Also, they state they wanted to remove VOCs with high reactivity, while they kept highly reactive ethylene, propylene, and m/p-xylene. Also, the authors state that they wanted to keep source tracers. If they knew what the source tracers were why would they need to perform a PMF analysis in the end? Would this a priori assumption not already induce some bias per se? Also, why did the authors not include acetylene, a well-known combustion/vehicle exhaust marker (e.g. Leuchner and Rappenglueck, 2010). The authors state that those selected 16 VOCs contributed > 75% of the total NMHCs together. What basis did they use for this

percentage, i.e. concentration, mixing ratio, ppbC or something else?

Page 5, L154-157: I do not agree on this (see my comments above on Text S3).

Page 5, L168: "The contribution of species...". Contribution to what?

Page 7, L211-216: This is not conclusive as long as biogenic VOCs, foremost isoprene are not considered.

Page 7, L226-229: I am not completely convinced about those statements on the T/B ratio. This ratio could also have changed due increased traffic emissions and/or changes in the traffic fleet (gasoline vs diesel, for instance). From Table S4 I see that in fact toluene decreased more than benzene over those years. However, I also see that xylenes - also typical tracers for solvents as toluene, - decreased way less than toluene or benzene.

Page 8, L236-257: This entire section is entirely based on the assumption that acetylene can be used as "...a marker for urban emissions". I doubt this. As mentioned earlier acetylene is a great tracer for combustion processes. It would not necessarily show up in appreciable amounts in other "urban emissions" like LPG or evaporation (again see for instance Leuchner and Rappenglueck, 2010).

Page 8, L264: I do not completely agree about the statement of little month-to-month variations; November and December data significantly stick out.

Page 8, L267-268: I do not agree. Propane has about the same kOH and ethane actually has a kOH which is about four times less than for benzene.

Page 8, L269-272: I thought the VOC data was corrected for meteorology. Why is there a dependence on temperature and wind? Also, if there is "heavy pollution" why would this only show up in three VOCs? In this case absolute values should be shown along with a definition what the authors would consider "heavy pollution".

Page 9, L273-274: What would be those "more indicative species"?

Page 9, L280-283: Again, with regard to petrochemical industry I suggest that the authors refer to Leuchner and Rappenglueck (2010).

Page 9, L286-287: Again, I thought the VOC data was corrected for meteorology.

Page 11, L349-352: This is not supported by the data shown in the paper, as Figure S5 does show changes and important VOCs like isoprene were not considered.

Figures S3: Datasets cannot be efficient. Also, what does the percentage exactly refer to?

Table S4: Are those VOC mixing ratios arithmetic means or medians? It would be helpful to include percentiles as well.

Table S6, last column: Contributions to what?

Reference:

Leuchner M. and Rappenglueck B. (2010): VOC Source-Receptor Relationships in Houston during TexAQS-II, Atmos. Environ., 44, 4056-4067, doi:10.1016/j.atmosenv.2009.02.029
* * *

---

## Referee Comment (RC2) · Anonymous Referee #3 · 13 Apr 2021

Preprint: acp-2020-1108

Title: Long term measurements (2009-2015) of non-methane hydrocarbons (NMHCs) in a megacity of China: implication for emission validation and source control

Authors: Peng et al.

The study of Peng et al. aimed to investigate the characteristics and sources of non-methane hydrocarbons (NMHCs) in Shanghai, China. The study was based on the

2009-2015 NMHC dataset, volatile organic compounds ratio, and positive matrix factorization (PMF)-derived factor temporal variation and trend analyses. The papers based on temporal variations, trend analysis, PMF, or toluene-to-benzene ratio are present in the literature for many years. The whole manuscript, the applied methods for data analysis, discussion, and conclusions are too generic and too basic. I would expect a more advanced methodological approach for revealing factors governing NMHCs' environmental fate, the evolution of their sources and sinks, or their interrelations with meteorological conditions. Moreover, I feel that the scientific novelty is missing.

Two of the three main objectives of this study (lines 67-69) were to "assess the source evolutions of NMHCs over time" and to "validate the speciation of emissions inventory primarily", but the authors used basic and straightforward methodology I believe is not capable of achieving them. I'll describe my concerns in the following text. Namely, regarding the applied methodology and concept, I have found some major shortcomings:

1. The authors excluded the influence of meteorological parameters: "we performed MLR model with ambient NMHCs and meteorological variables (temperature, wind speed and direction, air pressure and relative humidity) based on stepwise multiple linear regression". The first point I would like to address is the restriction to just five meteorological parameters. If the aim was to assess the impact of meteorology, the meteorological context had to be described broadly by using some of the available modeled data, i.e., Global Data Assimilation System with more than 20 important parameters. 2. The authors used 16 species for the source apportionment. Why the authors didn't use some of the standard procedures to determine the number of PMF-derived factors such as the scree plot (line 260)? Why the authors restrict the study to five? 3. 2009-2015 is a long period. If NMHCs concentration varied, why the authors didn't apply PMF for shorter periods, i.e., for each year separately (the data set is large enough), and try to see if a new source emerged or the composition of a particular source changed over time? This could happen, having in mind the influence of mitigation measures, technology development, or easily excluded meteorological

condition change (line 28) during 2009-2015. The analysis applied in this manuscript covered only PMF-derived factor trend and time variations, but not the variations in their composition which I think must have been included. 4. The authors argue that their primarily PMF-based analysis points out the omissions of emission source inventories (lines 35-36), which is the argument that I cannot agree with. To find the omissions in emission source inventories, I believe, a significantly advanced methodology/research has to be conducted with a disproportionately larger number of environmental factors included than the factors available in this study. There are many methods capable of modeling complex, heterogeneous, noisy, nonlinear, interactive, etc. interrelations between environmental factors such as machine learning (i.e. extreme gradient boosting). Moreover, there are many explainable artificial intelligence methods capable of explaining the derived dependencies in an extremely complex urban environment (i.e. Shapley additive explanations). PMF is not capable of meeting the goals of this study as the authors claim.

Proofreading by a native speaker is mandatory. It will clear some sentences, statements, and grammatical issues. I didn't make the corrections throw-out the manuscript because proofreading could significantly improve it.

---

## Author Comment (AC1) · 15 Jul 2021

**Responses to Reviewer's comment (acp-2020-1108)**

We thank the reviewers for the constructive and insightful comments, which significantly improved the quality of this work. Our point-by-point responses can be found below, with reviewer comments in **black**, our responses in **blue**, alongside the relevant revisions to the manuscript in red.

**Response to Anonymous Referee #2:**

The authors present an analysis of long-term VOC measurements in Shanghai. The main results are that ambient mixing ratios of alkanes showed an increasing trend, while aromatics and acetylene decreased. Alkenes did not reveal a trend. According to a PMF analysis emissions from vehicles, petrochemical industry and solvent use decreased, while emissions of natural gas and evaporation increased. Those results appear to be non-consistent with the emission inventory. In terms of O3 formation the authors state that decreasing VOC concentration would yield lower O3 forming potential, even more important for Shanghai as O3 formation would be in a VOC limited regime in that area.

When I first saw the paper I found it interesting, but when I read the supporting text materials my concerns grew and I had the impression that the conclusions the authors were drawing were not always well justified. I provide my detailed comments below. Apart from that I have been frustrated at times to see citations in the text without listing appropriate references, which is just bad. The use of English language in this paper is often poor and sometimes difficult to decode. It needs polishing.

1. Page 2, L28: "... of which composition varied significantly". I only saw that ethane and alkenes changed significantly (see Fig. 1 and associated discussion).

A:   The concentrations and interannual trends of major groups of non-methane hydrocarbons (NMHCs) were presented in Table R1. Except alkenes, three other groups including alkanes, aromatics and acetylene changed statistically ($p < 0.05$). The significant changes of compositions of NMHCs have been provided in the section 3.2.1 and Figure 3 in the revised manuscript.

Table R1. Arithmetic mean mixing ratios and trends of major NMHC groups and species in urban Shanghai during 2009-2015

| Species | Mixing ratio, ppbv | Trend, ppbv/yr | Annual Change Rate, /yr | $R^2$ | Abundance, % |
|---------|--------------------|----------------|-------------------------|-------|--------------|
| Alkanes | 13.6±1.7 | 0.51 | 3.8% | 0.40* | 52.6±5.1 |
| Alkenes | 3.2±0.6 | -0.05 | -1.3% | 0.03 | 12.2±1.4 |
| Aromatics | 6.9±1.2 | -0.50 | -6.0% | 0.70* | 27.0±4.2 |
| Acetylene | 2.1±0.4 | -0.24 | -7.8% | 0.72* | 8.2±1.3 |

The p-value is represented as follows: $p < 0.05$*.

2. Page 2, L28-30: This statement cannot be done as long as biogenic VOCs, foremost isoprene are not considered.

A: Isoprene was not included in this study. The reported mixing ratios of isoprene in Shanghai were summarized in Table R2. As reported, the mixing ratios of isoprene were in the range of 0.04–0.24 ppbv in NMHCs in urban Shanghai, and accordingly its ozone formation potential (OFP) was relatively low (0.60–3.61 ppbv). In this study, the total OFP of measured NMHCs was ~120 ppbv, and isoprene would account 2.3% at most with reported mixing ratio. Thus, the main OFP contributors in urban Shanghai were from anthropogenic emissions, even considering isoprene.

Table R2. Arithmetic mean mixing ratios and abundance of isoprene in Shanghai in previous studies

| References | Time | Mean mixing ratio/ppbv | Fraction of total measured NMHCs | Site |
|---|---|---|---|---|
| (Cai et al., 2010a) | 2006-2010 | 0.13±0.14 | 0.31% | urban |
|  |  | 0.06±0.06 | 0.21% | suburban |
|  |  | 0.07±0.09 | 0.38% | suburban |
| (Cai et al., 2010b) | 2007-2010 | 0.12±0.09 | 0.37% | urban |
| (Geng et al., 2010) | 2006-2008 | 0.24±0.14 | 0.50% | urban |
| (Wang et al., 2013) | 2009-2010 | 0.08 | 0.30% | urban |
| (Zhang et al., 2018) | 2013-2014 | 0.04±0.21 | 0.04% | suburban |
| (Liu et al., 2019) | 2017.5 | 0.13 | 0.55% | urban |

3. Page 3 L72-77: At some point the authors say the measurement site was 15 m above the ground, then at some point they say the site was at a five-story building. I somehow assume the site was on the rooftop of that building. However, I am not so much interested in how many stories a building would have, but I would rather be interested in what height the building would have and - assuming the site was on the rooftop - what the height of the sampling inlet above the rooftop was, and whether the rooftop had any outlet ducts. I neither saw the citation of Wang et al., 2016 nor Liu et al., 2019 in the list of references. What Wang et al., 2020 paper are the authors referring to, as in the list of references there is Wang et al., 2020a and Wang et al., 2020b....? The last sentence of this paragraph is very generic and vague: what do the authors consider "at a large enough temporal scale"?

A: Briefly, the sampling site was on the roof of a five-story building (~15 m above ground). The sampling inlet is around 1.5 m above the rooftop, and there are no other outlet ducts. We have rephrased the paragraph to avoid potential confusion.

"Online measurements of NMHCs were conducted continuously from 2009 to 2015 on the

roof of a 15 m height building in the campus of Shanghai Academy of Environmental Science (SAES, 31.17 N, 121.43 E). The sampling inlet is around 1.5 m above the rooftop, and there are no other outlet ducts."

We sincerely apologize for wrong citations, and we have updated the citations.

We have deleted the last sentence.

4. Page 3 L79: There have been many auto-GCs in use in the world for decades. Why is this GC in particular a "high-performance" instrument?

A: We have deleted it in the revised manuscript.

5. Page 4, L81-82: Still, I want to know briefly some standard information about the specifications of the traps, their temperature during preconcentration, the duration of pre concentration, the GC columns and the carrier gas used. Also, what sampling line (material, diameter, length) was used? Chromatograms for each channel should be shown.

A: Briefly, for C2–C6 hydrocarbons, the air sample was preconcentrated through a trap, a fine tube containing porous substances including Carbotrap C, Carbopack B and Carboxen. This trap was cooled at -8 °C for 4 min, then was heated at 220 °C. The total cycle duration was 30 min. Then, the C2–C6 hydrocarbons were separated on an ultimetal column (PLOT $Al_2O_3/Na_2SO_4$, 25 m length × 0.53 mm internal diameter × 10 μm film thickness), and detected by a flame ionization detector (FID).

For C6–C12 hydrocarbons, the air sample was preconcentrated through a trap with the trap phase (Carbotrap B), then the thermal desorption was fixed to 380°C, followed by separation on an ultimetal column (MXT 30 CE, 30 m × 0.28 mm × 1 μm) and also detected by a FID. The total cycle duration was 30 min. The detailed descriptions of the sampling procedure and instrument parameters can also be found elsewhere (Wang et al., 2013).

The ambient air was drawn through a teflon tube (OD:1/4") with the length of 1.5 m. The example of the chromatograms for two channels was shown in Figure R1.

We have revised the reference and clarified it more clearly in the revised manuscript.

"The ambient air was drawn through a teflon tube (OD:1/4") with the length of 1.5 m. For C2–C6 hydrocarbons, the air sample was preconcentrated through a trap, a fine tube containing porous substances including Carbotrap C, Carbopack B and Carboxen. This trap was cooled at -8 °C for 4 min, then was heated at 220 °C. The total cycle duration was 30 min. Then, the C2–C6 hydrocarbons were separated on an ultimetal column (PLOT $Al_2O_3/Na_2SO_4$, 25 m length × 0.53 mm internal diameter × 10 μm film thickness), and detected by a FID. The C6–C12 hydrocarbons were preconcentrated through a trap with the trap phase (Carbotrap B),

then the thermal desorption was fixed to 380℃, followed by separation on an ultimetal column (30 m × 0.28 mm × 1 μm) and also detected by a FID. The time resolution of the instrument was 30 min."

[Figure]

Figure R1. The Chromatograms for (a) C2–C6 hydrocarbons, (b) C6–C12 hydrocarbons on 14:00 Feb. 2nd in 2010.

6. Page 4 L83-84: Here the authors refer to Text S1. So I add my comments about Text S1 here as well. What specific VOCs were in the calibration gas? At what concentration were the single-point calibrations done? Apart from that it is not correct that using single-point calibration ensures the stability of the instrument. I do not understand the last sentence in the first paragraph ("Then...later years"). The authors state "...concentrations of most species measured agreed well with standard gas". How do the authors know this when actually the standard gas is used for calibration? The uncertainty should be calculated for each VOC individually and not for a group of VOCs.

A: The VOC Standards U.S. EPA PAMS mixture (provided by Spectra Gases Inc., USA) was applied in the calibration gas, then specific VOCs and the uncertainty were shown in Table R3 (also in Table S2). The measurement uncertainty of each species was estimated by the method in Text S1.

Table R3. Specific VOCs and the uncertainty of each VOC in the calibration gas

| Species | Uncertainty | Species | Uncertainty | Species | Uncertainty |
|---|---|---|---|---|---|
| Ethane | 18% | 2,3-Dimethylbutane | 18% | n-Octane | 6% |
| Ethylene | 12% | 2-Methylpentane | 20% | Ethylbenzene | 7% |
| Propane | 8% | 3-Methylpentane | 17% | m,p-Xylene | 16% |
| Propylene | 10% | n-Hexane | 13% | Styrene | 12% |
| Isobutane | 15% | 2-Methyl-1-pentene | 20% | o-Xylene | 14% |
| n-Butane | 17% | 1,3-Butadiene | 13% | n-Nonane | 7% |
| Acetylene | 13% | 2,4-Dimethylpentane | 18% | iso-Propylbenzene | 6% |
| trans-2-Butene | 17% | Benzene | 6% | n-Propylbenzene | 7% |
| 1-Butene | 14% | Cyclohexane | 6% | m-Ethyltoluene | 9% |
| cis-2-Butene | 16% | 2-Methylhexane/2,3-Dimethylpentane | 12% | p-Ethyltoluene | 11% |
| Cyclopentane | 13% | 3-Methylhexane | 6% | 1,3,5-Trimethylbenzene | 13% |
| Isopentane | 13% | 2,2,4-Trimethylpentane | 7% | o-Ethyltoluene | 8% |
| n-Pentane | 15% | n-Heptane | 7% | 1,2,4-Trimethylbenzene | 10% |
| trans-2-Pentene | 16% | Methylcyclohexane | 6% | n-Decane | 13% |
| 1-Pentene | 17% | 2,3,4-Trimethylpentane | 6% | 1,2,3-Trimethylbenzene | 11% |
| cis-2-Pentene | 16% | Toluene | 7% | m-Diethylbenzene | 11% |
| 2,2-Dimethylbutane | 19% | 2-Methylheptane | 6% | p-Diethylbenzene | 15% |
| Methylcyclopentane | 15% | 3-Methylheptane | 6% | | |

The single-point calibration was performed by using the internal permeation tubes with standard compounds every 24 h, which was used to evaluate the stability of the instrument performance. The permeation rates of standard materials (n-Hexane and Benzene) were 7.58 ng/min at 40 °C and 11.9 ng/min at 40 °C, respectively. Correspondingly, the mixing ratio of n-Hexane and Benzene in the gas flow was 8.28 ppb and 14.38 ppb at 25 °C, respectively. As shown in Figure R2, the single-point calibration results of n-Hexane and benzene were relatively stable except those during maintenance, which suggested the performance of the instrument was stable over the seven years measurements.

The multi-point calibration by standard gas was performed periodically with the points of 0.0 ppbv, 0.5 ppbv, 1.0 ppbv, 2.0 ppbv, and 4.0 ppbv. The measured results by GC-FID were used to compare with the reference mixing ratios of the standard gas diluted by dynamic gas calibrator. The measured results and the reference mixing ratios in standard gas had strong correlations ($R^2$=0.949–0.999). The target NMHCs was quantified using relative response factors being of the slopes of multi-point calibration standard curves (Table S2).

As shown in Figure R2 (a), the average of regression slopes between the measured and the reference concentrations in standard gas were stable over seven years. The several fluctuations of the daily single-point calibrations (Figure R2 (b) and (c)), were mainly caused by maintenances, such as replacement of permeation tubes, cold trap repair and parameter adjustment after repair, over the long-term period.

[Figure]

Figure R2. (a) The average of regression slopes between the measured and the reference concentrations in standard gases in multi-point calibrations. Error bar means one standard

deviation of the regression slopes of various species; (b) The relative variations of daily single point calibration with n-Hexane for C2~C6 monitoring system; (c) The relative variations of daily single point calibration with benzene for C6~C10 monitoring system. a, represents replacement of permeation tubes (on 8th Mar 2012, 29th Mar 2013, 22nd Mar 2014, 6th Nov 2014, respectively), b, represents column replacement on 1st Jun 2015, and c, represents repair of instrument during 18th Oct 2013–31st Dec 2013.

Overall, the stability and reliability of data was ensured by both multi-point and single-point calibrations. We have clarified it more clearly in the revised supplement (Text S1) and manuscript.

"Data quality assurance and control was ensured by both multi-point and single-point calibrations, and was described comprehensively in Text S1. Briefly, the fifteen times multi-point calibrations with standard gas (U.S. EPA PAMS mixture provided by Spectra Gases Inc., USA) were found to be repeatable over seven years (Figure S2). The single-point calibration was performed by using the internal permeation tubes with standard compounds every 24 h, which was used to evaluate the stability of the instrument performance."

As for the last sentence in Text S1, we now reword the text as "Next we clarified the data quality during 2009–2015 in detail." Additionally, the uncertainty of individual NMHC was shown in Table R3 (expressed as the accuracy in Table S2).

7. Page 4, L86-87: I cannot see in Table 2 that the responses of most species agree well with the standard. In fact many VOCs show slopes well below 1, in particular alkenes, foremost 1,3 butadiene, which is well-known for potential losses in GC preconcentration.

A: The target NMHC species was quantified using relative response factors being of the slopes of multi-point external standard curves (Table S2). The measured results and the reference concentrations in standard gas had strong correlations. That is, potential losses have been corrected with the response factors, when calculating the mixing ratios of individual NMHC by the regression slopes. We have clarified it more clearly in the revised manuscript.

"The target NMHCs was quantified using relative response factors being of the slopes of multi-point external standard curves (Table S2), where potential losses have been corrected."

8. Page 4, L88: Data cannot be "efficient".

A: We have revised it in the revised manuscript.

"Since the GC-FID was operated as an on-line mode, after excluding the missing data caused by replacement of permeation tubes, calibration runs, repair of instrument, unexpected shutdowns due to power outage, etc., all observed data were averaged as daily mixing ratios, with data coverage of 81.3% over seven years."

9. Page 4, L91-93: How can a trap be "out of work" for acetylene, while it is still properly working for all the remaining VOCs for an entire year? While it is true that acetylene shows challenges in GC analyses, I find it very astonishing/inconsistent that according to table S2 acetylene shows pretty good values for precision and accuracy. At this point I cannot follow at all the authors' statement in L91-93.

A: Acetylene is one of the most challenging compounds to measure. This is probably due to the concentration efficiency of acetylene in the trap is affected by water vapor more easily than other species. In this study, the trap was out of work for concentrating acetylene in 2015. There was no signal in the chromatogram, and the specifically technical reason was still unknown.

Actually, the acetylene showed pretty good measurements during 2009–2014. According to multi-point calibrations as shown in Figure R3, the performance of the instrument for acetylene analysis was stable (Slope=0.77±0.10, $R^2$=0.993).

[Figure]

Figure R3. The slopes and correlation coefficients of acetylene from multi-point calibrations during 2009–2014.

We now added the text in the revised manuscript.

"This was probably due to the concentration efficiency of acetylene in the trap, which was affected by water vapor more easily than other species. The specifically technical reason was still unknown."

10. Page 4, L95-105: Here the authors refer to Text S2 and Text S3. So I add my comments about Text S2 and Text S3 here as well. Text S2: I understand that the VOC data in the analysis of this paper has been corrected for meteorology. So why then can the authors make statements about evaporation processes (which depends on temperature) and regional transport (which would depend on wind) later on in the text, e.g. as shown in Figure S6 and other places in the paper?

A: In the present study, a stepwise multiple linear regression model (MLR) model was developed to test the meteorology effect on NMHCs interannual variability. The evaporation processes (which depends on temperature) and regional transport (which would depend on wind) referred in the paper and in Figure S6 indicated the monthly variations of NMHCs concentrations and sources, which could not be excluded by the MLR analysis focusing on the interannual variability.

11. Text S3: Equation 3-2 was derived for a rural area and in their paper Ehhalt and Rohrer (2000) point out that the coefficients should only be used for that specific rural airmass composition. Why do the authors think that equation 3-2 would be applicable the same way for the highly polluted area of the megacity Shanghai? I do not agree with the authors that Figure S5 shows insignificant trends of chemical loss for NMHCs over seven years. Just looking into L-Alkanes I would assume an overall increase by an order of 30% or so, while L-Alkenes show an overall decrease by about 25%. Also, L-Aromatics and L-Acetylene show high interannual variability, albeit no visible long-term trend. Apart from that I find it quite surprising to see an overall positive trend in OH concentrations (I would guess on the order of 30-50%), which the authors do not comment on at all.

A: The coefficients in equation 3-2, are obtained by a fit to the full set of 2124 measurements in a remote, rural area in POPCORN campaign (Ehhalt and Rohrer, 2000). The coefficients depend also on the ranges of the other variables influencing OH, such as the concentration of CO, which was small during POPCORN. We confirmed that there existed some uncertainty of the calculated OH radical concentration by equation 3-2 in Shanghai urban, which, however, was difficult to estimate due to lack of measurements of OH radicals. Zheng et al. (2011) used equation 3-2 to estimate the OH radical concentration in Beijing suburban and primarily estimated the uncertainty of the OH calculation was within 48% according to the measurements. Here, we assumed the uncertainty of the estimated OH radical concentrations was 48% in the present study, considering both Shanghai and Beijing were polluted areas relative to that in POPCORN campaign. It should be pointed that the present study focused on the long-term trend of NMHCs concentrations and sources as well as the potential impact of meteorology and photochemistry. Thus, the absolute OH radical concentrations estimated by equation 3-2 did exist large uncertainty which might be little for the trend of OH radical concentrations.

Thus, OH radical concentration ([OH] in molecule $cm^{-3}$) was estimated from the empirical equation (Ehhalt and Rohrer, 2000):

$$[OH]=4.1\times10^9\times(J_{O1D})^{0.83}\times(J_{NO2})^{0.19}\times\frac{140\times[NO_2]+1}{0.41\times[NO_2]^2+1.7\times[NO_2]+1} \quad (3\text{-}2)$$

During the long-term period, we only got ultraviolet A (UVA) data but not $J_{O1D}$ and $J_{NO2}$. Based on the fitting between UVA and $J_{O1D}$, $J_{NO2}$ which were measured in Shanghai in May of 2017, we scaled the observed UVA to get $J_{O1D}$ and $J_{NO2}$, as shown in Figure R4.

[Figure]

Figure R4. Fitting between the UVA and $J_{O1D}$, $J_{NO2}$ simultaneously measured in Shanghai in May of 2017.

Based on above fitting equation, the OH concentrations and related loss of NMHCs were showed in Figure R5 (we are sorry that the wrong version of figures was presented in the original manuscript). The OH concentration and chemical loss rate of NMHCs presented no significant changes based on statistic tests ($p > 0.05$), as listed in Table R4.

[Figure]

Figure R5. The annual mean loss rate of NMHCs and major groups by reaction with OH radical (red) and ozone (blue), respectively. L-NMHCs means chemical loss rate of NMHCs,

L-Alkanes, L-Alkenes, L-Aromatics, L-Acetylene represent the chemical loss rate of alkanes, alkenes, aromatics and acetylene, respectively. Error bars of OH concentration represented its uncertainty of calculation (48%) based on the results in Beijing suburban in China (Zheng et al., 2011). Error bars of ozone concentrations indicated one standard deviation from daily values.

Table R4. Arithmetic mean values and trends of [OH] (molecule $cm^{-3}$) and chemical loss rate of major NMHC groups (ppbv $s^{-1}$) in urban Shanghai during 2009-2015

| Name | Arithmetic mean value | Standard deviation | Trend,/yr | Annual Change Rate, /yr | $R^{2*}$ |
|---|---|---|---|---|---|
| [OH] | 9.28E+05 | 2.81E+04 | 7.34E+03 | 0.82% | 0.32 |
| L-NMHCs | 2.26E-04 | 3.22E-05 | -4.81E-06 | -1.79% | 0.10 |
| L-Alkanes | 3.25E-05 | 3.28E-06 | 6.15E-08 | 0.18% | 0.00 |
| L-Alkenes | 1.23E-04 | 2.40E-05 | -1.25E-06 | -0.81% | 0.01 |
| L-Aromatics | 6.87E-05 | 1.14E-05 | -3.53E-06 | -4.50% | 0.45 |
| L-Acetylene | 1.68E-06 | 3.31E-07 | -8.97E-08 | -3.93% | 0.34 |

*The p-values were larger than 0.05.

We have revised it in the revised supplement (Text S4).

12. The sentence "While the oxidation of NO3...neglected in this study" is irrelevant as this study did not include biogenic VOCs, anyway.

A: We have deleted it in the revised supplement.

13. The comparison analysis in the last paragraph does not mean anything as the "whole day" data would obviously include nighttime data as well.

A: The nighttime data independent of photochemistry was used to identify whether there are still significant trends of observed data, without the effect of photochemistry. This further indicated the significant trends of NMHCs levels were originated mainly from anthropogenic emissions rather than the photochemistry effect.

We have deleted it in the revised supplement.

14. Page 4, L108-112: I do not understand a couple of things here: In the first place, I do not see an appropriate definition of uncertainty for individual VOCs here. Anyway, assuming accuracy as shown in Table S2 as a proxy for uncertainty I do not understand why the authors kept VOCs with appreciable uncertainty for the PMF analyses (e.g. ethane, i-butane, n-butane, n-pentane, m/p-xylene) while they stated they wanted to remove VOCs with high uncertainty. Also, they state they wanted to remove VOCs with high reactivity, while they kept highly reactive ethylene, propylene, and m/p-xylene. Also, the authors state that they wanted to keep

source tracers. If they knew what the source tracers were why would they need to perform a PMF analysis in the end? Would this a priori assumption not already induce some bias per se? Also, why did the authors not include acetylene, a well-known combustion/vehicle exhaust marker (e.g. Leuchner and Rappenglueck, 2010). The authors state that those selected 16 VOCs contributed > 75% of the total NMHCs together. What basis did they use for this percentage, i.e. concentration, mixing ratio, ppbC or something else?

A: The uncertainty here indicated the measurement uncertainty of each NMHC species. In the positive matrix factorization (PMF) analysis, both the concentration and the measurement uncertainty of each species should be input into the PMF model. In the present study, the measurement uncertainty of each species was identified as following. For the species with the averaged measured concentration higher than the method detection limit (MDL) as listed in Table S2, their measurement uncertainty (unc) was calculated using the equation of $Unc=\sqrt{(Error\ fraction\times concentration)^2+(0.5\times MDL)^2}$. In comparison, for the species with measure concentration lower than the MDL, their measurement uncertainty was assumed as $unc=\frac{5}{6}\times MDL$.

In this study, 53 species were identified and quantified, 16 of which were applied into the PMF analysis. Generally, the relatively reactive species or species with concentration lower than MDL were given low priority to be input in the PMF analysis. An exception to this principle was the inclusion of species that are important tracers of sources. In the present study, the selection of species for the input of the PMF model was based on the following principles: (1) species with high abundance were usually selected, such as C2–C5 alkanes, ethylene, propene, toluene, xylenes and etc. Ethylene, propene, toluene and xylenes were selected not only due to their high abundance in Shanghai ambient, but also due to they are well known tracers of emissions. Specifically, ethylene and propene were considered as typical tracers from vehicle exhaust and petrochemical industries, and toluene and xylenes were well known tracers of solvent usages (Liu et al., 2019;Leuchner and Rappenglück, 2010), which were useful in source identification. (2) species with the average concentrations lower than the MDL were generally excluded, with an exception of styrene which was a good tracer of petrochemical industry. (3) highly reactive species, i.e., butenes and pentenes with lifetime of a few hours, were excluded although their concentrations were higher than the MDL. Acetylene was not included in the PMF analysis due to the lack of data in 2015. Eventually, 16 NMHC species were input into the PMF model to explore the sources of measured NMHCs. These species accounted for more than 75% (by ppbv) of the total mixing ratios of the measured NMHCs.

The PMF model is a multivariate factor analysis tool that decomposes a matrix of speciated sample data into two matrices including factor contributions and factor profiles (Norris et al., 2014). It can be interpreted as to what sources are represented based on observations at the receptor site. The special source tracers help us identify the resolved factors from PMF analysis to some extent. However, one NMHC may be emitted from several

sources in ambient. For instance, toluene could come from both solvent usage and traffic emissions. Hence, we cannot simply identify and quantify the sources variation from the change of tracers.

We have clarified it more clearly in the revised manuscript.

15. Page 5, L154-157: I do not agree on this (see my comments above on Text S3).

A: Please see our answer to question 11 above. We have revised it in the revised manuscript.

"The photochemistry effect on NMHCs trends was also tested in this study (Text S4), which was negligible (Figures S4-S5 and Table S6)."

16. Page 5, L168: "The contribution of species...". Contribution to what?

A: The contribution of species to total observed NMHCs mixing ratios in previous studies was summarized. We have revised the statement in Table S7 and the revised manuscript.

"The contribution of same species to total observed NMHCs mixing ratios in previous studies was summarized in Table S7, with the range of 79%–100%, indicating little bias with different species measured in previous studies."

17. Page 7, L211-216: This is not conclusive as long as biogenic VOCs, foremost isoprene are not considered.

A: Please see our response to question 2 above. The mixing ratios of isoprene were measured to be relatively low (<1% of measured NMHCs) in urban Shanghai, as reported. So, we considered the current ozone formation in urban Shanghai was dominated by anthropogenic NMHCs. We have clarified it in the revised manuscript.

18. Page 7, L226-229: I am not completely convinced about those statements on the T/B ratio. This ratio could also have changed due increased traffic emissions and/or changes in the traffic fleet (gasoline vs diesel, for instance). From Table S4 I see that in fact toluene decreased more than benzene over those years. However, I also see that xylenes - also typical tracers for solvents as toluene, - decreased way less than toluene or benzene.

A: Benzene has been prohibited in solvents, and it mainly come from combustion processes

especially fuel combustion from traffic emissions in megacities. In comparison, both the combustion and the industrial processes emitted toluene. Thus, the ratio of toluene to benzene (T/B) has been commonly used to identify the relative contribution of combustion versus industry emissions to some extent.

The decreasing rate of toluene (-8.3%/yr) annual concentrations was larger than that of benzene (-6.0%/yr), which resulted in the decrease of T/B ratio over seven years. It suggested both industrial and traffic emissions in Shanghai megacity have decreased in the past years, and specifically the decreasing rate of industrial emissions might be larger than that of traffic emissions. This could be true that the population of vehicles increased by 12% from 2009 to 2015, which to some extent offset the emission reduction from elimination of old vehicles.

As mentioned by the reviewer, xylenes, also typical tracers of solvent usage as toluene, had a lower decreasing rate than that of toluene, which might be because these two species were not always from the same emissions due to the complexity of industrial emissions.

We have stressed this in the revised manuscript.

19. Page 8, L236-257: This entire section is entirely based on the assumption that acetylene can be used as "...a marker for urban emissions". I doubt this. As mentioned earlier acetylene is a great tracer for combustion processes. It would not necessarily show up in appreciable amounts in other "urban emissions" like LPG or evaporation (again see for instance Leuchner and Rappenglueck, 2010).

A:  The emission ratio (ER) is the ratio between two species in their emissions, and generally the reference species is selected mainly based on the correlations between the reference species and other species. It suggested that most of NMHCs emissions should relate to acetylene emissions. In Shanghai, LPG usage was around 2.9%±0.5% in annual resident energy consumption per capita according to Shanghai Statistical Yearbook (2009–2015)(SMSB), which made little contribution to NMHCs emissions. As mentioned, acetylene could not be well associated with evaporation source. In the present study, most species showed significant correlations with measured acetylene, with the averaged correlation coefficient ranged from 0.5 to 0.8 (Table R5).

Table R5. The averaged correlation coefficient of species versus acetylene from orthogonal distance regression (ODR) fits.

| Species | R | Species | R | Species | R |
|---|---|---|---|---|---|
| Ethane | 0.74 | 2-Methylpentane | 0.56 | n-Octane | 0.54 |
| Ethylene | 0.71 | 3-Methylpentane | 0.56 | Ethylbenzene | 0.74 |
| Propane | 0.81 | n-Hexane | 0.48 | m,p-Xylene | 0.74 |
| Propylene | 0.54 | 2-Methyl-1-pentene | 0.29 | Styrene | 0.65 |
| Isobutane | 0.78 | 1,3-Butadiene | 0.43 | o-Xylene | 0.72 |
| n-Butane | 0.68 | 2,4-Dimethylpentane | 0.35 | n-Nonane | 0.67 |
| trans-2-Butene | 0.53 | Benzene | 0.75 | iso-Propylbenzene | 0.17 |
| 1-Butene | 0.56 | Cyclohexane | 0.63 | n-Propylbenzene | 0.55 |
| cis-2-Butene | 0.57 | 2-Methylhexane/2,3-Dimethylpentane | 0.58 | m-Ethyltoluene | 0.65 |

| | | | | | |
|---|---|---|---|---|---|
| Cyclopentane | 0.53 | 3-Methylhexane | 0.50 | p-Ethyltoluene | 0.63 |
| Isopentane | 0.52 | 2,2,4-Trimethylpentane | 0.44 | 1,3,5-Trimethylbenzene | 0.60 |
| n-Pentane | 0.60 | n-Heptane | 0.50 | o-Ethyltoluene | 0.61 |
| trans-2-Pentene | 0.42 | Methylcyclohexane | 0.56 | 1,2,4-Trimethylbenzene | 0.67 |
| 1-Pentene | 0.51 | 2,3,4-Trimethylpentane | 0.53 | n-Decane | 0.36 |
| cis-2-Pentene | 0.39 | Toluene | 0.68 | 1,2,3-Trimethylbenzene | 0.58 |
| 2,2-Dimethylbutane | 0.58 | 2-Methylheptane | 0.55 | m-Diethylbenzene | 0.53 |
| Methylcyclopentane | 0.55 | 3-Methylheptane | 0.54 | p-Diethylbenzene | 0.64 |
| 2,3-Dimethylbutane | 0.50 | | | | |

We provided the above discussions in the revised manuscript.

"(1) the reference species is selected mainly based on the correlations between the reference species and other species. It suggested that most of NMHCs emissions should relate to acetylene emissions. Acetylene has often been used as a marker for urban emissions (Boynard et al., 2014;Chen et al., 2016;Warneke et al., 2007). In the present study, most species showed significant correlations with measured acetylene, with the averaged correlation coefficient ranged from 0.5 to 0.8"

20. Page 8, L264: I do not completely agree about the statement of little month-to-month variations; November and December data significantly stick out.

A:   We have deleted it in the revised manuscript.

21. Page 8, L267-268: I do not agree. Propane has about the same kOH and ethane actually has a kOH which is about four times less than for benzene.

A:   This sentence has been revised in the revised manuscript.

"Furthermore, factor 2 also gave significant contribution to benzene which has a relatively long atmospheric lifetime, comparable with that of propane"

22. Page 8, L269-272: I thought the VOC data was corrected for meteorology. Why is there a dependence on temperature and wind? Also, if there is "heavy pollution" why would this only show up in three VOCs? In this case absolute values should be shown along with a definition what the authors would consider "heavy pollution".

A:   As mentioned above, in the present study, MLR model was conducted to test the

meteorology effect on NMHCs interannual variability. As shown in Figure 1 in the original manuscript, the interannual trends of NMHCs and the individual groups were independent of the meteorology effect. The evaporation processes (which depends on temperature) and regional transport (which mainly depends on wind speed and wind direction) referred in the paper and in Figure S6 indicated the monthly variations of NMHCs concentrations and sources, which could not be excluded by the MLR analysis focusing on the interannual variability.

We defined the factor 2 as the mixture of natural gases and regional transport. Generally, NMHCs were atmospheric consumed gradually during the regional transport, in which the relatively unreactive species like benzene, ethane and propane with long lifetime took up larger amount than the other reactive species.

The heavy pollution here was representative of the worse air quality in northwest of Shanghai in cool seasons. Under the prevailing wind in winter, relatively aged air masses and unreactive VOC species may be transported into Shanghai. The term "heavy pollution" used may indeed lead to a misinterpretation, and we have changed our wording in the revised manuscript.

23. Page 9, L273-274: What would be those "more indicative species"?

A: More indicative species like methane which is the main component of natural gas might be a more suitable indicator of natural gas emissions. While, the level of methane in ambient is higher by ~two orders of those of NMHCs, which also a challenge to be used as the indicator of natural gas in the source apportionment of NMHCs by PMF model. More studies were essential to apportion these two kinds of sources in future.

24. Page 9, L280-283: Again, with regard to petrochemical industry I suggest that the authors refer to Leuchner and Rappenglueck (2010).

A: We have added the reference in the revised manuscript.

25. Page 9, L286-287: Again, I thought the VOC data was corrected for meteorology.

A: Please see the response of question 10 above. The evaporation processes (which depends on temperature) referred in the paper and in Figure S6 indicated the monthly variations of NMHCs concentrations and sources, which could not be excluded by the MLR analysis focusing on the interannual variability.

26. Page 11, L349-352: This is not supported by the data shown in the paper, as Figure S5 does show changes and important VOCs like isoprene were not considered.

A: Please see the response of question 2 and 11 above. There was no significant long-term trend on calculated loss for NMHCs based on statistic tests. Isoprene took up 0.04–0.55% of the total NMHCs in Shanghai, which could not impact the long-term trend of NMHCs concentrations in Shanghai.

27. Figures S3: Datasets cannot be efficient. Also, what does the percentage exactly refer to?

A: We have updated the statement in the revised Figure. The percentage showed here refers to the proportion of days with available data on each month.

[Figure]

Figure R6: Data availability of observed NMHCs during 2009–2015

28. Table S4: Are those VOC mixing ratios arithmetic means or medians? It would be helpful to include percentiles as well.

A: The mixing ratios are arithmetic means of annual averages of seven years. The results in the table are calculated based on annual averages of seven years, and thus the percentiles are meaningless. We have clarified it more clearly in supplement.

29. Table S6, last column: Contributions to what?

A: The contributions of same species with our study, to the total mixing ratios of measured species. We have revised it in the supplement.

**Response to Anonymous Referee #3:**

The study of Peng et al. aimed to investigate the characteristics and sources of nonmethane hydrocarbons (NMHCs) in Shanghai, China. The study was based on the 2009-2015 NMHC dataset, volatile organic compounds ratio, and positive matrix factorization (PMF)-derived factor temporal variation and trend analyses. The papers based on temporal variations, trend analysis, PMF, or toluene-to-benzene ratio are present in the literature for many years. The whole manuscript, the applied methods for data analysis, discussion, and conclusions are too generic and too basic. I would expect a more advanced methodological approach for revealing factors governing NMHCs' environmental fate, the evolution of their sources and sinks, or their interrelations with meteorological conditions. Moreover, I feel that the scientific novelty is missing.

A: With the implementation of a series of clean air action in China, declines in many air pollutants both in ambient concentration and emissions were reported widely, while, worsened surfaced ozone pollution occurred meantime (Ma et al., 2016;Sun et al., 2016;Xu et al., 2019;Shen et al., 2019) . The reported increasing emissions of anthropogenic NMHCs in China (Li et al., 2019c) was recognized as an important role in the worse of ozone pollution, which was attributed to the absence of effective controlling measures of NMHCs. It seemed not consistent with the reality.

The present study, taking Shanghai megacity as an example, comprehensively investigated the long term NMHCs measurements as well as the social and economic detailed activities data to explore the evolution of emissions as well as the effectiveness of air pollution control measures in China in the past. The present study provided evidence of the inconsistency between observations (decreasing) and emission inventories (increasing), both in interannual trend and speciation as well as source contributions, emphasizing the need for further validation in NMHCs emission inventory in future. What's more, the assessment of source-by-source trend suggested differential effect of the past controlling measures in China. To our most knowledge, the present study was the only work resulted the statistic decrease of NMHCs concentrations and differential trends of their emissions in a megacity of China on an interannual scale. We resulted the effectiveness of controlling measures for mobile and centralized emissions from large industrial sources in the past in contrast to those for fugitive emissions, which provided new insights into future clean air policies in polluted region. From this point of view, we believed the present study was deserved to be considered for publication.

The method employed in this study including mathematic statistics for trend analysis of NMHCs concentrations and sources, PMF analysis for source apportionment, and multiple linear regression model (MLR) model to filter the meteorology effect. These methods were basic but still useful for atmospheric community, which actually did provide many valuable results of the evolution of emissions as well as the effectiveness of air pollution control measures in China in the past. The suggested advanced methods like machine learning might

be useful for the purpose of the present study but we cannot have detailed enough relevant input datasets of emissions and activity.

Two of the three main objectives of this study (lines 67-69) were to "assess the source evolutions of NMHCs over time" and to "validate the speciation of emissions inventory primarily", but the authors used basic and straightforward methodology I believe is not capable of achieving them. I'll describe my concerns in the following text. Namely, regarding the applied methodology and concept, I have found some major shortcomings:

1. The authors excluded the influence of meteorological parameters: "we performed MLR model with ambient NMHCs and meteorological variables (temperature, wind speed and direction, air pressure and relative humidity) based on stepwise multiple linear regression". The first point I would like to address is the restriction to just five meteorological parameters. If the aim was to assess the impact of meteorology, the meteorological context had to be described broadly by using some of the available modeled data, i.e., Global Data Assimilation System with more than 20 important parameters.

A: We updated the MLR analysis by using the Global Data Assimilation System in the revised manuscript as following:

In hundreds of variables in the European Centre for Medium-Range Weather Forecasts (ECMWF) reanalysis dataset, 10 candidate variables were selected as they are observed to be correlated with compositions and levels of NMHCs (Li et al., 2019a;Thera et al., 2019;Liu et al., 2017) or air pollutants (Zhai et al., 2019;Li et al., 2019b;Otero et al., 2018) (Table R6).We reanalyzed the correlation of NMHCs with 10 candidate meteorological variables from reanalysis databases (ERA5 hourly data, accessible at https://cds.climate.copernicus.eu/cdsapp#!/search?type=dataset&text=ERA5).

Table R6. Meteorological variables used in the regression models [a].

| Variable Name | Meteorological Parameters Description |
| --- | --- |
| PBLH | Boundary layer height (m) |
| D | 2m dewpoint temperature  (K) |
| SLP | Mean sea level pressure  (Pa) |
| Tmax | Maximum 2m temperature since previous post-processing (K) |
| SSR | Surface net solar radiation ($J\ m^{-2}$) |
| T | 2m temperature (K) |
| TCC | Total cloud cover (%) |
| TP | Total precipitation (m) |
| U10 | 10m u-component of wind ($m\ s^{-1}$) |
| V10 | 10m v-component of wind ($m\ s^{-1}$) |

ª All meteorological variables are 24-h averages. Except for total precipitation and cloud cover, all data are from ERA5 reanalysis dataset with spatial resolution of $0.25° \times 0.25°$.

MLR models have been successfully applied to quantify the effect of meteorological variability on $PM_{2.5}$ and ozone in elsewhere (Tai et al., 2010;Li et al., 2019b). Similarly, we apply the MLR model with ambient NMHCs and major meteorological variables during the long-term period, to rule out the interference from meteorology on interannual variability of NMHCs.

Specifically, all data including grouped NMHCs and meteorological variables were deseasonalized and detrended by substracting the 12-month moving averages from the original monthly data. This focused the correlations on synoptic time scales, avoiding aliasing from common seasonal variations or long-term trends (Tai et al., 2010;Zhai et al., 2019). The model is of the form (Zhai et al., 2019):

$$Y_d(t) = \sum_{k=1}^{10} \beta_k X_{d,k}(t) + b \qquad (2\text{-}1)$$

$Y_d(t)$ represents the deseasonalized and detrended grouped NMHCs time series and $X_{d,k}(t)$ is the corresponding time series for the meteorological factors. We fit the regression coefficients $\beta_k$ and the intercept b with stepwise linear regression, by adding or deleting terms based on their independent statistical significance to obtain the best model fit (Zhai et al., 2019;Tai et al., 2010). The regression coefficient $\beta_k$ is zero for meteorological variable not included in the final MLR model. The variance inflation factor (Velleman and Welsch, 1981) ranges between 1.0 and 1.8, indicating that the problem of multicollinearity among meteorological variables is generally unimportant. Finally, the best fit of grouped NMHCs ($\beta_k$)was found:

a. NMHC=-0.992+4.412U10

b. ALKANE=-0.171+1.296U10-6.422E-6SSR-0.01TCC

c. ALKENE=-0.218+0.61U10-0.002PBLH

d. AROMATIC=-0.249+1.774U10+0.112Tmax

e. ACETYLENE=-0.141+0.434U10+0.000361SLP     (2-2)

Then the NMHCs anomaly $Y_a$ is obtained by deseasonalizing but not detrending the NMHCs data by removing the 7-year means for that month of the year, as well as the meteorological anomalies $X_{a,k}$. The meteorology-driven NMHCs anomalies $Y_m$:

$$Y_m(t) = \sum_{k=1}^{10} \beta_k X_{a,k}(t) + b \qquad (2\text{-}3)$$

After removing the meteorological influence from the MLR model, the residual anomaly $Y_r$ :

$$Y_r(t) = Y_a(t) - Y_m(t) \qquad\qquad (2\text{-}4)$$

After removing meteorological influence, the residual (meteorology-corrected data) may include noise due to limitations of the MLR model, but the long-term trend over the 7-year period could be reasonably attributed to the effect of anthropogenic emissions changes.

In the original manuscript, the meteorology effect was estimated based on 5 measured meteorological variables (Temperature, Relative Humidity, Pressure, Wind Direction and Wind Speed) using the MLR models. As shown in Figure R7, the results of two dataset (the reanalysis data and measurements) used in MLR models were very similar.

In the revised manuscript, we have updated Figure 1 with ERA5 reanalysis meteorology data.

[Figure]

Figure R7. The trends of monthly mixing ratios of NMHCs and the major chemical categories (ppbv) from observation (in black) and meteorology-corrected data (the meteorological variables from ERA5 data (in red), and from measurements (in blue)) during 2009–2015. Average annual change rate (ppbv/% yr⁻¹) was derived from yearly values.

We have updated the statement in revised manuscript and supplement (Text S2).

"A stepwise multiple linear regression (MLR) model was developed to rule out the interference from meteorology on interannual variability of NMHCs. MLR models have been successfully applied to quantify the effect of meteorological variability on $PM_{2.5}$ and ozone in elsewhere (Tai et al., 2010;Otero et al., 2018;Zhai et al., 2019;Li et al., 2019c). In hundreds of variables in the European Centre for Medium-Range Weather Forecasts (ECMWF) reanalysis dataset (ERA5), 10 candidate variables were selected as they are observed to be correlated with compositions and levels of NMHCs (Li et al., 2019a;Thera et al., 2019;Liu et al., 2017) or air pollutants (Zhai et al., 2019;Li et al., 2019c;Otero et al., 2018).We reanalyzed the correlation of NMHCs with 10 candidate meteorological variables from reanalysis databases (ERA5 hourly data) over the long-term period. The model fit the deseasonalized and detrended monthly NMHCs time series to the related meteorological variables with stepwise regression, by adding or deleting terms based on their independent statistical significance. Then meteorology-driven anomalies considered to be excluded, resulting the meteorology-corrected values (Details in Text S2). After removing meteorological influence, the long-term trends over the 7-year period could be reasonably attributed to the effect of anthropogenic emissions changes"

2. The authors used 16 species for the source apportionment. Why the authors didn't use some of the standard procedures to determine the number of PMF-derived factors such as the scree plot (line 260)? Why the authors restrict the study to five?

A: According to the PMF, an observed concentration at a receptor site can be viewed as a data matrix **X** of $i$ by $j$ dimensions in which i number of samples and j number of measured species (Paatero and Tapper, 1994). The object function Q, based on the uncertainties inherent in each observation, can allow the analyst to review the distribution for each species to evaluate the stability of the solution:

$$Q = \sum_{i=1}^{n} \sum_{j=1}^{m} \frac{e_{ij}^2}{s_{ij}^2} = \sum_{i=1}^{n} \sum_{j=1}^{m} \left( \frac{x_{ij} - \sum_{k=1}^{p} g_{ik} f_{kj}}{s_{ij}} \right)^2 \quad (1)$$

Where $g_{ik}$ is the mass contribution of the $k$th source to the $i$th sample, $f_{kj}$ is the $j$th species mass fraction from the $k$th source, $p$ is the number of possible source factors, and $e_{ij}$ is the residual associated with the concentration of the $j$th species in the $i$th sample, $g_{ik} \geq 0$ and $f_{kj} \geq 0$ and where $n$ is the number of samples, $m$ is the number of considered species, and $s_{ij}$ is an uncertainty for the $j$th species measured in the $i$th sample. Normally, monitoring the total Q is not meaningful because the expected value for a suitable solution (Qexp) depends on the size of the data matrix and on the number of chosen factors. The Q/Qexp values can be used to choose the reasonableness of the model results, and a value of ~1 generally indicates the relatively reasonable solution. The calculation of Qexp is shown in following Equation:

$$Q_{exp} = n \times m - p \times (n + m) \quad (2)$$

In this study, PMF model run ranging from 4 to 8 factor numbers were carried out to ascertain the most reasonable solution. Each simulation was randomly conducted 10 times.

The most appropriate number of factors (F value) was selected by some mathematical indicators calculated following the PMF mPodel, including the Q values, a possible explanation of the sources, and the residual distribution. Almost 100 % of the scaled residuals were within ±3σ and were normally distributed for all species. The Q/Qexpected value for F= 5 was closer than 1.0 (e.g., 1.11 in comparison with 1.34 for F= 4 and 0.88 for F= 6), suggesting that the five-factor configuration was supposed, as shown in Figure R8.

[Figure]

Figure R8. The Q/Qexpected values for different PMF solutions.

The final number of factor was determined based on both the mathematic and physical rationality. The resolved profiles of four, five and six PMF-derived factors were shown in Figure R9. The resolved result of 4-factors could not sperate vehicle exhaust with other factors to some extent. While for the result of 6-factor, the source of vehicle exhaust was split into two meaningless profiles. Thus, 5-factor PMF results were presented in the manuscript.

[Figure]

Figure R9. (A-C): Source profiles resolved from PMF model (blue bars), and distributions of each species among these factors (black dots), with 4–6 factors respectively.

The statement was added in the revised supplement and manuscript.

"In this analysis, PMF model runs ranging from 4 to 8 factor numbers were carried out to ascertain the most reasonable solution. Each simulation was randomly conducted 10 times. The most appropriate number of factors were selected by some mathematical indicators calculated following the PMF model, including the objective function (Q) values, a possible explanation of the sources, and the residual distribution. Almost 100 % of the scaled residuals were within ±3σ and were normally distributed for all species. The Q/Qexpected value for five-factor was closer to 1.0, suggesting that the five-factor configuration was supposed."

3. 2009-2015 is a long period. If NMHCs concentration varied, why the authors didn't apply PMF for shorter periods, i.e., for each year separately (the data set is large enough), and try to see if a new source emerged or the composition of a particular source changed over time? This could happen, having in mind the influence of mitigation measures, technology development, or easily excluded meteorological condition change (line 28) during 2009-2015. The analysis applied in this manuscript covered only PMF-derived factor trend and time variations, but not the variations in their composition which I think must have been included.

A: The source apportionment analysis has been separately studied for each year. 16 species of measured NMHCs were selected as the input of the PMF model in the present study. Generally, the relatively reactive species or species with concentration lower than MDL were given low priority to be input in the PMF analysis. An exception to this principle was the inclusion of species that are important tracers of sources. In the present study, the selection of species for the input of the PMF model was based on the following principles: (1) species with high abundance were usually selected, such as C2–C5 alkanes, ethylene, propene, toluene, xylenes and etc. Ethylene, propene, toluene and xylenes were selected not only due to their high abundance in Shanghai ambient, but also due to they are well known tracers of emissions. Specifically, ethylene and propene were considered as typical tracers from vehicle exhaust and petrochemical industries, and toluene and xylenes were well known tracers of solvent usages (Liu et al., 2019;Leuchner and Rappenglück, 2010), which were useful in source identification. (2) species with the average concentrations lower than the MDL were generally excluded, with an exception of styrene which was a good tracer of petrochemical industry. (3) highly reactive species, i.e., butenes and pentenes with lifetime of a few hours, were excluded although their concentrations were higher than the MDL. Acetylene was not included in the PMF analysis due to the lack of data in 2015. Eventually, 16 NMHC species were input into the PMF model to explore the sources of measured NMHCs. These species accounted for more than 70% (by ppbv) of the total mixing ratios of the measured NMHCs during 2009–2015 (Figure R10)

[Figure]

Figure R10. The proportions of selected species in total measured NMHCs in each year (% by ppbv). The dashed line is 70%.

Following the standard procedures of PMF, the Q/Qexpected value for each year were presented in Figure R11.

[Figure]

Figure R11. The Q/Qexpected values for different PMF solutions.

The factor number was determined based on both the mathematic and physical rationality. The resolved profiles of four, five and six PMF-derived factors were shown in Figure R12. The mean profiles from each year and multi-year results are similar based on statistic tests (the test statistic t was not significant). This was expected because the abundant species of NMHCs in each year were similar and the measured data were continuous. Our work aimed to investigate the trends of NMHCs sources in urban Shanghai. In order to minimize the potential impact of PMF solutions between different years (although not significant) in trends analysis, we presented the multi-year PMF results in this study. The compositions of the resolved sources have been provided in Figure 7(a) in the revised manuscript.

[Figure]

Figure R12. (A-C): Source profiles resolved from PMF model (bars), and distributions of each species among these factors (dots), with 4–6 factors respectively. Blue bars and black dots mean the values over seven years. Red bars and grey dots mean averaged results in each year separately, error bars represented one standard deviation of the results in each year.

The statement was added in the revised supplement and manuscript.

"Considering the number of sources could be various over the seven years, we performed the PMF analysis in each year and in 7-year together. As a result, the averaged source profiles from each year and multi-years results were similar based on statistic tests (the test statistic t was not significant) (Text S3, Figure S8). This was expected because the abundant species of NMHCs in each year were similar and the measured data were continuous. In order to minimize the potential impact of PMF solutions between different years (although not significant) in trends analysis, we presented the multi-years PMF results in this study."

4. The authors argue that their primarily PMF-based analysis points out the omissions of emission source inventories (lines 35-36), which is the argument that I cannot agree with. To find the omissions in emission source inventories, I believe, a significantly advanced methodology/research has to be conducted with a disproportionately larger number of environmental factors included than the factors available in this study. There are many methods capable of modeling complex, heterogeneous, noisy, nonlinear, interactive, etc. interrelations between environmental factors such as machine learning (i.e. extreme gradient boosting). Moreover, there are many explainable artificial intelligence methods capable of explaining the derived dependencies in an extremely complex urban environment (i.e. Shapley additive explanations). PMF is not capable of meeting the goals of this study as the authors claim.

A: A number of recent studies have demonstrated the application of machine learning algorithms to the understanding of the large and multivariate datasets typical of atmospheric sciences (Ivatt and Evans, 2020;Keller et al., 2021). For air pollutants, machine learning has been usually used for improving parameterization in climate models, specifically to ozone ($O_3$) and particulate matter ($PM_{2.5}/PM_{10}$), or for forecasts and predictions (Nowack et al., 2018;Watson et al., 2019). The data-driven machine learning focuses on the determinants of air pollutants to create formula for prediction. For example, random forest (RF), as one of the machine learning models, are used to analyze the air pollution. Generated by RF (Breiman, 2001), feature importance (FI) (Watson et al., 2019) is able to quantitatively rate the significance of each input to the output. Liu et al. (2020) and Lu et al. (2021) estimated $O_3$ and $PM_{2.5}$ concentrations via RF, respectively. They focus on the complicated relationship among all air pollutants and metrological conditions to straighten out which variables are more distinguished for an air pollutant. These methods mostly input detailed related parameters to analyze the variables we concerned. The suggested advanced methods like machine learning might be useful for the purpose of the present study but we cannot have any detailed relevant input datasets of emissions and activity of each NMHC species.

The present study, taking Shanghai megacity as an example, comprehensively investigated the long term NMHCs measurements as well as the social and economic detailed activities data to explore the evolution of emissions as well as the effectiveness of air pollution control measures in China in the past. The present study provided evidence of the

inconsistency between observations (decreasing) and emission inventories (increasing), both in interannual trend and speciation as well as source contributions, emphasizing the need for further validation in NMHCs emission inventory in future. What's more, the assessment of source-by-source trend suggested differential effect of the past measures in China.

The method employed in this study including mathematic statistics for trend analysis of NMHCs concentrations and sources, PMF analysis for source apportionment, and multiple linear regression model (MLR) model to filter the meteorology effect. These methods were basic but still useful for atmospheric community, which actually did provide many valuable results of the evolution of emissions as well as the effectiveness of air pollution control measures in China in the past.

Proofreading by a native speaker is mandatory. It will clear some sentences, statements, and grammatical issues. I didn't make the corrections throw-out the manuscript because proofreading could significantly improve it.

A:   The manuscript has been polished carefully.

References:

Breiman, L.: Random Forests, Machine Learning, 45, 5-32, 10.1023/A:1010933404324, 2001.

Cai, C. J., Geng, F. H., Tie, X. X., Yu, Q., Peng, L., and Zhou, G. Q.: Characteristics of ambient volatile organic compounds (VOCs) measured in Shanghai, China, Sensors (Basel), 10, 7843-7862, 10.3390/s100807843, 2010a.

Cai, C. J., Geng, F. H., Tie, X. X., Yu, Q. O., and An, J. L.: Characteristics and source apportionment of VOCs measured in Shanghai, China, Atmospheric Environment, 44, 5005-5014, 10.1016/j.atmosenv.2010.07.059, 2010b.

Ehhalt, D. H., and Rohrer, F.: Dependence of the OH concentration on solar UV, J Geophys Res-Atmos, 105, 3565-3571, Doi 10.1029/1999jd901070, 2000.

Geng, F., Cai, C., Tie, X., Yu, Q., An, J., Peng, L., Zhou, G., and Xu, J.: Analysis of VOC emissions using PCA/APCS receptor model at city of Shanghai, China, Journal of Atmospheric Chemistry, 62, 229-247, 10.1007/s10874-010-9150-5, 2010.

Guo, H., Cheng, H. R., Ling, Z. H., Louie, P. K., and Ayoko, G. A.: Which emission sources are responsible for the volatile organic compounds in the atmosphere of Pearl River Delta?, J Hazard Mater, 188, 116-124, 10.1016/j.jhazmat.2011.01.081, 2011.

Ivatt, P. D., and Evans, M. J.: Improving the prediction of an atmospheric chemistry transport model using gradient-boosted regression trees, Atmos Chem Phys, 20, 8063-8082, 10.5194/acp-20-8063-2020, 2020.

Keller, C. A., Evans, M. J., Knowland, K. E., Hasenkopf, C. A., Modekurty, S., Lucchesi, R. A., Oda, T., Franca, B. B., Mandarino, F. C., Suarez, M. V. D., Ryan, R. G., Fakes, L. H., and Pawson, S.: Global impact of COVID-19 restrictions on the surface concentrations of nitrogen dioxide and ozone, Atmos Chem Phys, 21, 3555-3592, 10.5194/acp-21-3555-2021, 2021.

Leuchner, M., and Rappenglück, B.: VOC source–receptor relationships in Houston during TexAQS-II, Atmospheric Environment, 44, 4056-4067, 10.1016/j.atmosenv.2009.02.029, 2010.

Li, B. W., Ho, S. S. H., Gong, S. L., Ni, J. W., Li, H. R., Han, L. Y., Yang, Y., Qi, Y. J., and Zhao, D. X.: Characterization of VOCs and their related atmospheric processes in a central Chinese city during severe ozone pollution periods, Atmos Chem Phys, 19, 617-638, 10.5194/acp-19-617-2019, 2019a.

Li, K., Jacob, D. J., Liao, H., Shen, L., Zhang, Q., and Bates, K. H.: Anthropogenic drivers of 2013-2017 trends in summer surface ozone in China, Proc Natl Acad Sci U S A, 116, 422-427, 10.1073/pnas.1812168116, 2019b.

Li, M., Zhang, Q., Zheng, B., Tong, D., Lei, Y., Liu, F., Hong, C. P., Kang, S. C., Yan, L., Zhang, Y. X., Bo, Y., Su, H., Cheng, Y. F., and He, K. B.: Persistent growth of anthropogenic non-methane volatile organic compound (NMVOC) emissions in China during 1990-2017: drivers, speciation and ozone formation potential, Atmos Chem Phys, 19, 8897-8913, 10.5194/acp-19-8897-2019, 2019c.

Liu, C. T., Ma, Z. B., Mu, Y. J., Liu, J. F., Zhang, C. L., Zhang, Y. Y., Liu, P. F., and Zhang, H. X.: The levels, variation characteristics, and sources of atmospheric non-methane hydrocarbon compounds during wintertime in Beijing, China, Atmos Chem Phys, 17, 10633-10649, 10.5194/acp-17-10633-2017, 2017.

Liu, H., Liu, J., Liu, Y., Ouyang, B., Xiang, S., Yi, K., and Tao, S.: Analysis of wintertime O3 variability using a random forest model and high-frequency observations in Zhangjiakou-an area with background pollution level of the North China Plain, Environ Pollut, 262, 114191, 10.1016/j.envpol.2020.114191, 2020.

Liu, Y., Wang, H., Jing, S., Gao, Y., Peng, Y., Lou, S., Cheng, T., Tao, S., Li, L., Li, Y., Huang, D., Wang, Q., and An, J.: Characteristics and sources of volatile organic compounds (VOCs) in Shanghai during summer: Implications of regional transport, Atmospheric Environment, 215, 10.1016/j.atmosenv.2019.116902, 2019.

Lu, J., Zhang, Y. H., Chen, M. X., Wang, L., Zhao, S. H., Pu, X., and Chen, X. G.: Estimation of monthly 1 km resolution PM2.5 concentrations using a random forest model over "2+26" cities, China, Urban Clim, 35, 10.1016/j.uclim.2020.100734, 2021.

Ma, Z. Q., Xu, J., Quan, W. J., Zhang, Z. Y., Lin, W. L., and Xu, X. B.: Significant increase of surface ozone at a rural site, north of eastern China, Atmos Chem Phys, 16, 3969-3977, DOI 10.5194/acp-16-3969-2016, 2016.

Norris, G., R. Duvall, S. Brown, and Bai, S.: EPA Positive Matrix Factorization (PMF) 5.0

Fundamentals and User Guide, U.S. Environmental Protection Agency, Washington, DC, Report EPA/600/R-14/108 (NTIS PB2015-105147), 2014.

Nowack, P., Braesicke, P., Haigh, J., Abraham, N. L., Pyle, J., and Voulgarakis, A.: Using machine learning to build temperature-based ozone parameterizations for climate sensitivity simulations, Environ Res Lett, 13, 10.1088/1748-9326/aae2be, 2018.

Otero, N., Sillmann, J., Mar, K. A., Rust, H. W., Solberg, S., Andersson, C., Engardt, M., Bergstrom, R., Bessagnet, B., Colette, A., Couvidat, F., Cuvelier, C., Tsyro, S., Fagerli, H., Schaap, M., Manders, A., Mircea, M., Briganti, G., Cappelletti, A., Adani, M., D'Isidoro, M., Pay, M. T., Theobald, M., Vivanco, M. G., Wind, P., Ojha, N., Raffort, V., and Butler, T.: A multi-model comparison of meteorological drivers of surface ozone over Europe, Atmos Chem Phys, 18, 12269-12288, 10.5194/acp-18-12269-2018, 2018.

Paatero, P., and Tapper, U.: Positive Matrix Factorization - a Nonnegative Factor Model with Optimal Utilization of Error-Estimates of Data Values, Environmetrics, 5, 111-126, DOI 10.1002/env.3170050203, 1994.

Shen, L., Jacob, D. J., Liu, X., Huang, G. Y., Li, K., Liao, H., and Wang, T.: An evaluation of the ability of the Ozone Monitoring Instrument (OMI) to observe boundary layer ozone pollution across China: application to 2005-2017 ozone trends, Atmos Chem Phys, 19, 6551-6560, 10.5194/acp-19-6551-2019, 2019.

SMSB: Shanghai Municipal Statistics Bureau: Shanghai Statistical Yearbook (2009–2015), China Statistics Press, Beijing, China,

Sun, L., Xue, L. K., Wang, T., Gao, J., Ding, A. J., Cooper, O. R., Lin, M. Y., Xu, P. J., Wang, Z., Wang, X. F., Wen, L., Zhu, Y. H., Chen, T. S., Yang, L. X., Wang, Y., Chen, J. M., and Wang, W. X.: Significant increase of summertime ozone at Mount Tai in Central Eastern China, Atmos Chem Phys, 16, 10637-10650, 10.5194/acp-16-10637-2016, 2016.

Tai, A. P. K., Mickley, L. J., and Jacob, D. J.: Correlations between fine particulate matter (PM2.5) and meteorological variables in the United States: Implications for the sensitivity of PM2.5 to climate change, Atmospheric Environment, 44, 3976-3984, 10.1016/j.atmosenv.2010.06.060, 2010.

Thera, B. T. P., Dominutti, P., Öztürk, F., Salameh, T., Sauvage, S., Afif, C., Çetin, B., Gaimoz, C., Keleş, M., Evan, S., and Borbon, A.: Composition and variability of gaseous organic pollution in the port megacity of Istanbul: source attribution, emission ratios, and inventory evaluation, Atmos Chem Phys, 19, 15131-15156, 10.5194/acp-19-15131-2019, 2019.

Velleman, P. F., and Welsch, R. E.: Efficient Computing of Regression Diagnostics, The American Statistician, 35, 234-242, 10.1080/00031305.1981.10479362, 1981.

Wang, H. L., Chen, C. H., Wang, Q., Huang, C., Su, L. Y., Huang, H. Y., Lou, S. R., Zhou, M., Li, L., Qiao, L. P., and Wang, Y. H.: Chemical loss of volatile organic compounds and its impact on the source analysis through a two-year continuous measurement, Atmospheric Environment, 80, 488-498, 10.1016/j.atmosenv.2013.08.040, 2013.

Wang, H. L., Qiao, L. P., Lou, S. R., Zhou, M., Ding, A. J., Huang, H. Y., Chen, J. M., Wang,

Q., Tao, S., Chen, C. H., Li, L., and Huang, C.: Chemical composition of PM2.5 and meteorological impact among three years in urban Shanghai, China, J Clean Prod, 112, 1302-1311, 10.1016/j.jclepro.2015.04.099, 2016.

Wang, H. L., Yan, R. S., Xu, T. T., Wang, Y. H., Wang, Q., Zhang, T. Q., An, J. Y., Huang, C., Gao, Y. Q., Gao, Y., Li, X., Yu, C., Jing, S. G., Qiao, L. P., Lou, S. R., Tao, S. K., and Li, Y. J.: Observation Constrained Aromatic Emissions in Shanghai, China, J Geophys Res-Atmos, 125, 10.1029/2019JD031815, 2020.

Watson, G. L., Telesca, D., Reid, C. E., Pfister, G. G., and Jerrett, M.: Machine learning models accurately predict ozone exposure during wildfire events, Environ Pollut, 254, 112792, 10.1016/j.envpol.2019.06.088, 2019.

Xu, J. M., Tie, X. X., Gao, W., Lin, Y. F., and Fu, Q. Y.: Measurement and model analyses of the ozone variation during 2006 to 2015 and its response to emission change in megacity Shanghai, China, Atmos Chem Phys, 19, 9017-9035, 10.5194/acp-19-9017-2019, 2019.

Zhai, S., Jacob, D. J., Wang, X., Shen, L., Li, K., Zhang, Y., Gui, K., Zhao, T., and Liao, H.: Fine particulate matter (PM2.5) trends in China, 2013–2018: separating contributions from anthropogenic emissions and meteorology, Atmos Chem Phys, 19, 11031-11041, 10.5194/acp-19-11031-2019, 2019.

Zhang, Y. C., Li, R., Fu, H. B., Zhou, D., and Chen, J. M.: Observation and analysis of atmospheric volatile organic compounds in a typical petrochemical area in Yangtze River Delta, China, Journal of Environmental Sciences, 71, 233-248, 10.1016/j.jes.2018.05.027, 2018.

Zheng, J., Hu, M., Zhang, R., Yue, D., Wang, Z., Guo, S., Li, X., Bohn, B., Shao, M., He, L., Huang, X., Wiedensohler, A., and Zhu, T.: Measurements of gaseous H2SO4 by AP-ID-CIMS during CAREBeijing 2008 Campaign, Atmos. Chem. Phys., 11, 7755-7765, 10.5194/acp-11-7755-2011, 2011.